# Template switching during DNA replication is a prevalent source of adaptive gene amplification

Julie N Chuong[1], Nadav Ben Nun[2,3], Ina Suresh[1], Julia Cano Matthews[1], Titir De[1], Grace Avecilla[4], Farah Abdul-Rahman[5,6], Nathan Brandt[7], Yoav Ram[2,3], David Gresham[1]*

[1]Department of Biology, Center for Genomics and Systems Biology, New York University, New York, United States; [2]School of Zoology, Faculty of Life Sciences, Tel Aviv University, Tel Aviv, Israel; [3]Edmond J. Safra Center for Bioinformatics, Tel Aviv University, Tel Aviv, Israel; [4]Department of Natural Sciences, Baruch College CUNY, New York, United States; [5]Department of Ecology and Evolutionary Biology, Yale University, New Haven, United States; [6]Microbial Sciences Institute, Yale University, New Haven, United States; [7]Department of Biological Sciences, North Carolina State University, Raleigh, United States

*For correspondence:
dgresham@nyu.edu

Competing interest: The authors declare that no competing interests exist.

## eLife Assessment

This study provides **important** new insights into the contributions of local DNA features to the complex molecular mechanisms and dynamics of copy number variation (CNV) formation during adaptive evolution. While limited to a single CNV of interest, the study is well-designed and carefully controlled, presenting **compelling** evidence that supports the conclusions. This work will be of general interest to those studying genome architecture and evolution from yeast biologists to cancer researchers.

**Abstract** Copy number variants (CNVs) are an important source of genetic variation underlying rapid adaptation and genome evolution. Whereas point mutation rates vary with genomic location and local DNA features, the role of genome architecture in the formation and evolutionary dynamics of CNVs is poorly understood. Previously, we found the *GAP1* gene in *Saccharomyces cerevisiae* undergoes frequent amplification and selection in glutamine-limitation. The gene is flanked by two long terminal repeats (LTRs) and proximate to an origin of DNA replication (autonomously replicating sequence, ARS), which likely promote rapid *GAP1* CNV formation. To test the role of these genomic elements on CNV-mediated adaptive evolution, we evolved engineered strains lacking either the adjacent LTRs, ARS, or all elements in glutamine-limited chemostats. Using a CNV reporter system and neural network simulation-based inference (nnSBI) we quantified the formation rate and fitness effect of CNVs for each strain. Removal of local DNA elements significantly impacts the fitness effect of *GAP1* CNVs and the rate of adaptation. In 177 CNV lineages, across all four strains, between 26% and 80% of all *GAP1* CNVs are mediated by Origin Dependent Inverted Repeat Amplification (ODIRA) which results from template switching between the leading and lagging strand during DNA synthesis. In the absence of the local ARS, distal ones mediate CNV formation via ODIRA. In the absence of local LTRs, homologous recombination can mediate gene amplification following *de novo* retrotransposon events. Our study reveals that template switching during DNA replication is a prevalent source of adaptive CNVs.

## Introduction

Defining the genetic basis and evolutionary dynamics of adaptation is a central goal in evolutionary biology. Mutations underlying adaptation or biological innovation can depend on multiple factors including genetic backgrounds, phenotypic states, and genome architecture (*Blount et al., 2008*; *Blount et al., 2012*). One important class of mutation mediating adaptive evolution are copy number variants (CNVs) which comprise duplications or deletions of genomic sequences that range in size from gene fragments to whole chromosomes. Quantifying the rates at which CNVs occur, the factors that influence their formation, and the fitness and functional effects of CNVs is essential for understanding their role in evolutionary processes.

CNVs play roles in rapid adaptation in multiple contexts and are an initiating event in biological innovation. For example, in laboratory evolution experiments a spontaneous tandem duplication captured a promoter for expression of a citrate transporter and resulted in *Escherichia coli* cells, typically unable to use citrate, to start metabolizing citrate as a carbon source (*Blount et al., 2012*). CNVs can be beneficial in cancer cells, promote tumorigenesis (*Ben-David and Amon, 2020*), enhance cancer cell adaptability (*Rutledge et al., 2016*), and accelerate resistance to anti-cancer therapies (*Lukow et al., 2021*). Over longer time scales, CNVs serve as substrate from which new genes evolve (*Ohno, 1970*; *Taylor and Raes, 2004*) as duplicated genes redundant in function can accumulate mutations and evolve to acquire new functions. For example, the globin gene family in mammals arose from rounds of gene duplication and subsequent diversification (*Storz, 2016*). CNVs also contribute to macro-evolutionary processes and thereby contribute to species differences, such as between humans and chimpanzees (*Cheng et al., 2005*) and reproductive isolation (*Zuellig and Sweigart, 2018*).

Mutations, including CNVs, occur in part because of errors made during DNA replication or DNA repair. Two general processes underlie CNV formation: (1) DNA recombination-based mechanisms and (2) DNA replication-based mechanisms (*Brewer et al., 2011*; *Harel et al., 2015*; *Hastings et al., 2009a*; *Malhotra and Sebat, 2012*; *Pös et al., 2021*; *Zhang et al., 2009a*). Recombination-mediated mechanisms of CNV formation include non-allelic homologous recombination (NAHR) and nonhomologous end joining. NAHR occurs via recombination between homologous sequences that are not allelic. As such, NAHR occurs more frequently with repetitive sequences due to improper alignment of DNA segments and can occur either between (interchromosomal) or within (intrachromosomal) a chromosome (*Harel et al., 2015*). One prevalent class of repetitive sequence are retrotransposons and both full length and partial sequences, such as long terminal repeats (LTR), are substrates for homologous recombination generating gene amplifications (*Avecilla et al., 2023*; *Dunham et al., 2002*; *Gresham et al., 2008*; *Lauer et al., 2018*; *Spealman et al., 2022*). Separately, repetitive or palindromic DNA can form hairpin structures and thus are thought to be sites for CNV formation after double-strand DNA breakage (*Lobachev et al., 2002*; *Narayanan et al., 2006*). DNA replication-based mechanisms include fork stalling template switching (FoSTeS) and microhomology mediated break-induced replication (MMBIR; *Carvalho et al., 2013*; *Gu et al., 2008*; *Hastings et al., 2009b*; *Lee et al., 2007*). During FoSTeS and MMBIR, after a DNA replication fork stalls and a DNA replication error occurs in which the lagging strand switches to an incorrect template strand mediated by microhomology. Reinitiation of DNA synthesis at the incorrect site can form CNVs. A particular type of DNA replication-based error is Origin Dependent Inverted Repeat Amplification (ODIRA), in which short inverted repeats and an active origin of DNA replication enable template switching of the leading strand to the lagging strand template. Subsequent replication generates an intermediate DNA molecule that can recombine into the original genome to form a triplication with an inverted middle copy (*Brewer et al., 2011*; *Brewer et al., 2015*; *Martin et al., 2024*).

In microbes, CNVs can mediate rapid adaptation to selective conditions imposed through nutrient limitation in a chemostat. Selected CNVs often include genes encoding nutrient transporters that facilitate import of the limiting nutrient (*Dunham et al., 2002*; *Gresham et al., 2008*; *Hong and Gresham, 2014*; *Horiuchi et al., 1963*; *Payen et al., 2016*; *Sonti and Roth, 1989*), likely as a result of improved nutrient transport capacity due to increased protein production. Previous studies have found amplification of the general amino acid permease gene, *GAP1*, when *Saccharomyces cerevisiae* populations are continuously cultured in glutamine-limited chemostats (*Gresham et al., 2010*; *Lauer et al., 2018*). Amplification of *GAP1* confers increased fitness in the selective environment (*Avecilla et al., 2023*; *Lauer et al., 2018*). Sequence characterization of these CNVs revealed that a diversity of *de novo* CNV alleles are generated and selected including tandem duplications, complex large CNVs,

aneuploidies, and translocations. However, little is known about the molecular mechanisms underlying this diversity.

Local genome sequence elements are likely to be an important determinant of CNV formation rates and mechanisms. Genomic context can influence multiple properties including mutation rate, epigenetic regulation, chromatin state, transcription levels, DNA replication, and recombination rate (*Arndt et al., 2005*; *Chuang and Li, 2004*; *Lang and Murray, 2011*; *Lercher and Hurst, 2002*; *Matassi et al., 1999*; *Nishant et al., 2009*; *Wolfe et al., 1989*). Prior work has shown that CNVs occur more frequently in repetitive regions in the genome (*Harel et al., 2015*; *Pentao et al., 1992*; *Stankiewicz et al., 2003*; *Turner et al., 2008*). However, little is known about the role of local genomic architecture and organization on CNV formation rates, the types of CNVs that are generated, their associated fitness effects, and ultimately the paths taken during adaptive evolution.

Here, we aimed to investigate the effect of local genome architecture elements on *de novo GAP1* CNV formation and selection dynamics during adaptive evolution of *Saccharomyces cerevisiae*. We hypothesized that sequence elements proximate to *GAP1* potentiate CNV formation. The *GAP1* locus, which is located on the short arm of chromosome XI, consists of two flanking Ty1 long terminal repeats (LTRs) that share 82% sequence identity and an origin of DNA replication or autonomously replicating sequence (ARS; *Figure 1A*). Both LTRs and ARS may facilitate *GAP1* CNV formation due to their proximity. First, the flanking LTRs can undergo inter-chromatid NAHR to form tandem duplications of *GAP1* on a linear chromosome (*Lauer et al., 2018*; *Spealman et al., 2022*). Second, intra-chromatid NAHR between the flanking LTRs can form an extrachromosomal circle containing *GAP1* and an ARS able to self-propagate and integrate into the genome (*Gresham et al., 2010*). Finally, *GAP1* triplications can form through ODIRA using short inverted repeats and the proximate ARS (*Brewer et al., 2015*; *Lauer et al., 2018*; *Martin et al., 2024*). These elements are thought to facilitate a high rate of *GAP1* amplification, estimated to be on the order of $10^{-4}$ per haploid genome per generation (*Avecilla et al., 2022*). To test our hypothesis we used a CNV reporter, wherein a constitutively expressed fluorescent GFP gene is inserted adjacent to *GAP1* (*Lauer et al., 2018*). We engineered strains that lacked either the ARS (ARSΔ), both flanking LTRs (LTRΔ), or all three elements (ALLΔ; *Figure 1A*). We performed experimental evolution using wildtype (WT) and genomic architecture mutant populations in glutamine-limited chemostats for 137 generations and quantified *GAP1* CNVs using flow cytometry (*Figure 1*). Surprisingly, we find that the proximate DNA elements are not required for *GAP1* CNV formation as *GAP1* CNVs were identified in all evolving populations. We used neural network simulation-based inference (nnSBI) to infer the CNV formation rate and selection coefficient (*Avecilla et al., 2022*). We find that although genomic architecture mutants have significantly reduced CNV formation rates relative to WT and significantly lower selection coefficients, *GAP1* CNVs repeatedly form and sweep to high frequency in all strains with modified genomes. We performed genome sequence analysis to define the molecular mechanisms of CNV formation for 177 CNV lineages and found that 26–80% of *GAP1* CNVs are mediated by ODIRA across all four background strains. In the absence of the local ARS, a distal ARS facilitates CNV formation through ODIRA. We also find that homologous recombination mechanisms still mediate gene amplification in the absence of LTRs in part initiated by *de novo* insertion of retrotransposon elements at the locus. Our study reveals the remarkable plasticity of the genome and that template switching during DNA replication is a common source of adaptive CNVs even in the absence of the wildtype local DNA sequences.

## Results

Accurate estimation of CNV allele frequencies in heterogeneous populations remains challenging using molecular methods such as DNA sequencing and qPCR. To address this challenge we previously developed a CNV reporter comprising a constitutively expressed fluorescent gene inserted upstream of *GAP1* and observed recurrent amplification and selection of *GAP1* in glutamine-limiting chemostats (*Lauer et al., 2018*). Subsequently, we showed that a high rate of *GAP1* CNV formation and strong fitness effects explain the highly reproducible evolutionary dynamics (*Avecilla et al., 2022*). Noncoding sequence elements proximate to *GAP1*, including flanking LTRs in tandem orientation and an ARS, contribute to *GAP1* CNV formation (*Gresham et al., 2010*; *Lauer et al., 2018*). Many studies have shown that repetitive sequence regions and origins of replications are hotspots of CNVs (*Arlt et al., 2012*; *Cardoso et al., 2016*; *Di Rienzi et al., 2009*; *Gresham et al., 2010*; *Lauer et al., 2018*;

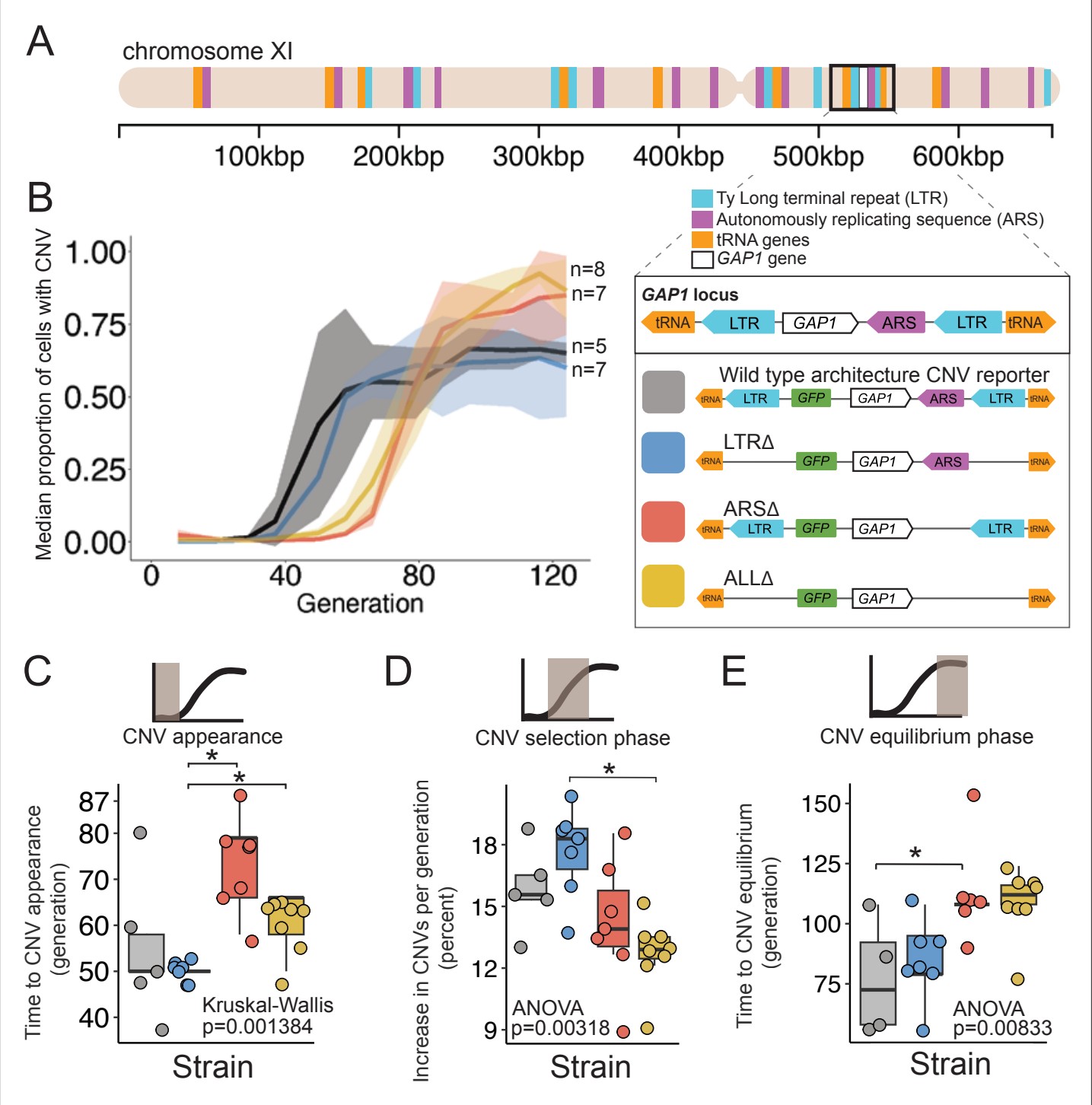

**Figure 1.** A local DNA replication origin contributes to CNV dynamics during adaptive evolution. (**A**) The *Saccharomyces cerevisiae GAP1* gene is located on the short arm of chromosome XI (beige rectangle). Light blue rectangle - Ty Long terminal repeats (LTR). Purple rectangle - Autonomously replicating sequences (ARS). Orange rectangles - tRNA genes. *GAP1* ORF - white rectangle. The *GAP1* gene (white rectangle) is flanked by Ty1 LTRs (YKRCδ11, YKRCδ12), which are remnants of retrotransposon events, and is directly upstream of an autonomously replicating sequence (ARS1116). Variants of the *GAP1* locus were engineered to remove either both LTRs, the single ARS, or all three elements. All engineered genomes contain a CNV reporter. (**B**) We evolved the four different strains in 5–8 replicate populations, for a total of 27 populations, in glutamine-limited chemostats and monitored the formation and selection of *de novo GAP1* CNVs for 137 generations using flow cytometry. Population samples were taken every 8–10 generations and 100,000 cells were assayed using a flow cytometer. Colored lines show the median proportion of cells in a population with *GAP1* amplifications across 5–8 replicate populations of the labeled strain. The shaded regions represent the median absolute deviation across the replicates.

*Figure 1 continued on next page*

Figure 1 continued

(**C**) We summarized CNV dynamics and found that strain has a significant effect on time to CNV appearance (Kruskal-Wallis, p=0.001384). There are significant differences in time to CNV appearance between LTRΔ (blue) and ARSΔ (red), and LTRΔ (blue) and ALLΔ (yellow) (pairwise wilcoxon test with Bonferroni correction, p=0.0059 and p=0.0124, respectively). (**D**) Strain has a significant effect on the per generation increase in proportion of cells with CNV (ANOVA, p=0.00318) calculated as the slope during CNV selection phase. There is a significant difference between LTRΔ (blue) and ALLΔ (yellow) (pairwise t-test with Bonferroni correction, p=0.0026). (**E**) Strain has a significant effect on time to CNV equilibrium (ANOVA, p=0.00833). There is a significant difference in time to CNV equilibrium between WT and ARSΔ (pairwise t-tests with bonferroni correction, p=0.050).

*Zhang et al., 2013*). Thus, we hypothesized that the local genomic architecture of *GAP1* facilitates its high rate of CNV formation.

To test the role of proximate genomic features we engineered strains in which each element is deleted and thus differ from the wildtype strain (WT) containing a *GAP1* CNV reporter by a single modification. Specifically, we constructed ARSΔ, a strain lacking the single ARS, LTRΔ, a strain lacking the flanking LTRs, and ALLΔ, a strain lacking all three elements (*Figure 1A*). All strains contain the CNV reporter at the identical location as the WT strain. We confirmed scarless deletions of genetic elements using Sanger and whole-genome sequencing.

## Local genomic architecture contributes to *GAP1* CNV evolutionary dynamics

We founded independent populations with each of the three engineered strains lacking proximate genomic features and a WT strain. We studied *GAP1* CNV dynamics in populations maintained in glutamine-limited chemostats over 137 generations (*Figure 1*). For each of the four strains, we propagated 5–8 clonal replicate populations, each originating from the same inoculum (founder population) derived from a single colony. Approximately every 10 generations, we measured GFP fluorescence of sampled populations using a flow cytometer and quantified the proportion of cells containing *GAP1* CNVs (Methods). We observed similar CNV dynamics across independent populations within each strain (*Figure 3—figure supplement 1*). Therefore, we summarized CNV dynamics for each strain using the median proportion of the population with a *GAP1* CNV (*Figure 1B*). In every strain, *GAP1* CNVs are generated and selected resulting in qualitatively similar dynamics in WT and mutant strains.

## Deletion of the ARS, but not the flanking LTRs alters CNV dynamics

We quantified three phases of CNV dynamics: (1) time to CNV appearance, defined by the inflection point before the rise in CNV proportion (*Figure 1C*); (2) selection of CNV, corresponding to the increase in proportion of CNVs per generation during the initial expansion of CNVs (i.e. slope; *Figure 1D*); and (3) equilibrium phase, corresponding to the inflection point before the plateau (*Figure 1E*). The time to CNV appearance (*Figure 1C*) and the CNV selection (*Figure 1B*) does not differ between WT and LTRΔ populations (pairwise Wilcoxon test, adjusted p=1, pairwise t-test adjusted p=1, respectively). In the WT and LTRΔ populations, *GAP1* CNVs appear at generation 50 (*Figure 1C*) and increase in proportion at similar rates, ~15% per generation in WT and ~18% per generation in LTRΔ (*Figure 1D*). The two strains both reach their equilibrium phase at the same time, around generation 75 (pairwise t-test, adjusted p=1; *Figure 1E*). The absence of a significant difference in CNV dynamics between the two strains suggests that the LTRs are not a major determinant of *GAP1* CNV evolutionary dynamics.

By contrast, in ARSΔ and ALLΔ populations, we observe a delay in the time to CNV appearance. In both of these strains, CNVs are first detected at generations 65–80, whereas in WT and LTRΔ populations CNVs are first detected at generation 50 (ARSΔ vs. LTRΔ, wilcoxon pairwise test, adjusted p=0.0059, ALLΔ vs. LTRΔ, Wilcoxon pairwise test, adjusted, p=0.0124; *Figure 1C*). Thus, the local ARS contributes to the initial *GAP1* CNV dynamics. Similarly, CNV selection is significantly different between the LTRΔ (18%) and ALLΔ (13%; pairwise t-test, adjusted p-value = 0.0026; *Figure 1D*). Finally, we also observe a significant delay (ANOVA, p=0.00833) in the generation at which the CNV proportion reaches equilibrium in ARSΔ (~generation 112) compared to WT (pairwise t-test, adjusted p=0.05; *Figure 1E*). These observations suggest that absence of the ARS in the ARSΔ and ALLΔ strains delays the appearance of *GAP1* CNVs compared with the presence of the ARS in WT or LTRΔ strains.

### *GAP1* amplifications can occur without CNV reporter amplification

In both WT and LTRΔ populations we observed that *GAP1* CNV abundance stabilized around 75% during the equilibrium phase (*Figure 1B*) across each of the 12 independent populations (*Figure 3— figure supplement 1*). Flow cytometry analysis showed that each experiment begins with a population of cells with only one-copy of GFP (*Figure 2A*). Over generations, distinct populations appear with higher GFP fluorescence (*Figure 2A*). Previously, GFP fluorescence has been shown to scale with *GAP1* copy number (*Lauer et al., 2018*). Therefore, the four distinct subpopulations observed (*Figure 2A*) likely represent cells harboring 1, 2, 3, and 4-copies of *GAP1,* respectively. This corroborates previous experimental evolution results in which *de novo* GAP1 CNVs are quickly formed and selected for and over selection higher copy outcompete lower copy subpopulations (*Lauer et al., 2018*). The raw flow cytometry plots (*Figure 2—source data 1*) and population GFP histograms (*Figure 2—source data 2*) also revealed a persistent single-copy GFP subpopulation throughout the timecourse (*Figure 2A,* bottom subpopulation in each panel). These data could be explained by two possible scenarios: (1) the existence of a non-*GAP1* CNV subpopulation comprising beneficial variation at other loci with fitness effects equivalent to *GAP1* CNVs; or (2) lineages with *GAP1* CNVs without co-amplification of the CNV reporter. To resolve these two possibilities, we sequenced clones from the single-copy GFP subpopulation across the five WT populations from different chemostats (*Supplementary file 1*) and identified the presence of *GAP1* amplifications without co-amplification of the CNV reporter in four out of five WT populations (*Figure 2—figure supplement 1*). We found eleven distinct *GAP1* CNVs that lacked amplification of the reporter gene (*Figure 2—figure supplement 1*) indicating at least eleven independent CNV events occurred, either in the founder population or shortly after chemo-stat inoculation. By contrast, in one of the five populations, population 3, all clones from the single-copy GFP subpopulation contained one copy of GFP and one copy of *GAP1* (*Supplementary file 1*), suggesting these clones have a beneficial mutation elsewhere in the genome that allows their stable coexistence with the *GAP1* CNV subpopulation. The *GAP1* CNVs without GFP amplification were either pre-existing at the time of the inoculation or occurred shortly after inoculation. We suspect that they likely occurred after inoculation but early in the evolution, for three reasons: (1) the similar ~75% plateau is observed in the dynamics in all independent WT and LTRΔ populations, (2) at least 1 inde-pendent CNVs of this type were detected (*GAP1* amplification without co-amplification of the GFP), and (3) there are no common CNVs detected across chemostats. Our findings show that *GAP1* amplifi-cation without coamplification of the CNV reporter can occur and beneficial variation other than *GAP1* CNVs underlie adaptation to glutamine-limitation.

### Incorporating unreported early-occurring CNVs in an evolutionary model

To quantify the evolutionary parameters underlying empirically measured CNV dynamics (*Figure 1*) we built a mathematical evolutionary model, which describes the experiment in a simplified manner. Because measuring CNV rates and selection coefficients is difficult and laborious to perform in the lab, we use neural network simulation-based inference (nnSBI) to estimate these parameters (*Avecilla et al., 2022*; *Cranmer et al., 2020*; *Gonçalves et al., 2020*). Additionally, we use the model to predict hypothetical outcomes of additional experiments without requiring additional experimentation. We have previously used nnSBI to infer *GAP1* CNV formation rates and selection coefficients in glutamine-limited selection and experimentally validated these inferences using barcode tracking and pairwise competition assays (*Avecilla et al., 2022*). Previously, our evolutionary model assumed the *GAP1* CNV reporter allowed us to detect all *GAP1* CNVs. However, our new flow cytometry and sequencing results indicate the existence of a small subpopulation of unreported *GAP1* CNVs present either at the beginning or early in the experiments. Therefore, we expanded the evolutionary model to include $\varphi$, the proportion of cells with *GAP1* CNVs without co-amplification of the reporter, at the commence-ment of the experiment (i.e. generation 0). The remaining model parameters are $\delta_C$, the rate at which *GAP1* duplications form; $\delta_B$, the rate other beneficial mutations occur; $s_C$, the selection coefficient of *GAP1* CNVs; and $s_B$, the fitness effect of other beneficial mutations (*Figure 2B*). We find that this expanded evolutionary model can accurately describe the observed dynamics (*Figure 3—figure supplement 1*), which are clearly affected by the value of $\varphi$. When the total CNV proportion is very different from the reported proportion, for example when $\varphi \gg \delta_C > \delta_B$, a reduced CNV formation rate results in a greater discrepancy between reported and total CNV proportions (*Figure 2C*).

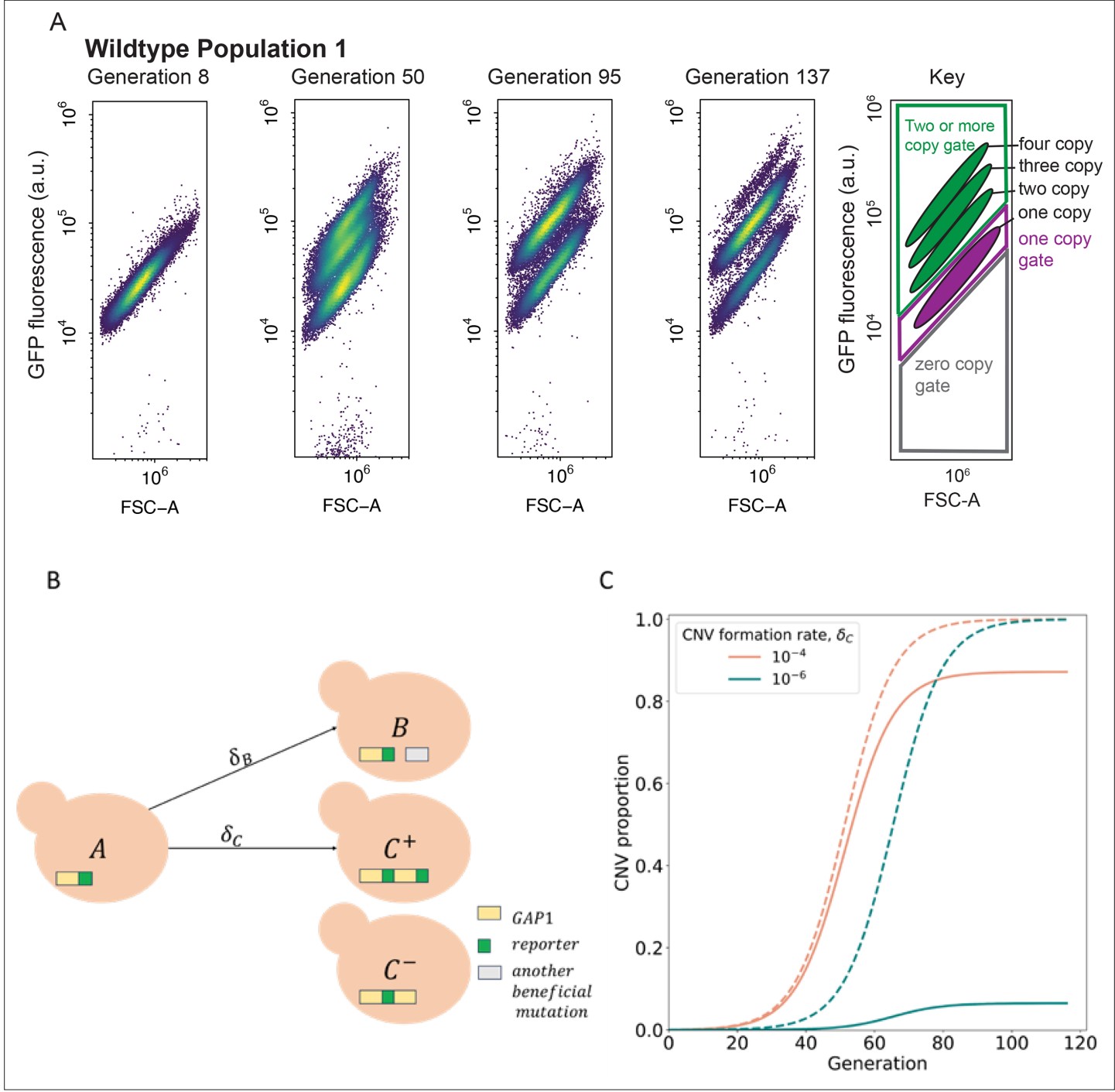

**Figure 2.** CNV reporter failure does not impact parameter inference. (**A**) Flow cytometry of a representative WT population with a persistent one-copy GFP subpopulation, bottom in each panel. FSC-A is forward scatter-area which is a proxy for cell size (x-axis). GFP fluorescence was measured in arbitrary units (a.u.) (y-axis). Hierarchical gating was performed to define the one-copy GFP and two-or-more copy subpopulations (see Methods). (**B**) Model illustration. $X_A$ is the frequency of ancestor cells in the chemostat; $X_{C+}$, $X_{C-}$ are the frequencies of cells with *GAP1* duplications with two or one reporters, respectively, and a selection coefficient $s_C$; $X_B$ is the frequency of cells with other beneficial mutations and a selection coefficient $s_B$. *GAP1* duplications form with a rate $\delta_C$, other beneficial mutations occur with rate $\delta_B$. At generation 0, only genotypes $C^-$ and $A$ are present, with frequencies of $X_{C-} = \varphi$ and $X_A = 1 - \varphi$. (**C**) Examples of total CNV proportions (dashed) and reported CNV proportions (solid) for two parameter combinations, both with $s_C = 0.15$, $\varphi = 10^{-4}$.

The online version of this article includes the following source data and figure supplement(s) for figure 2:

**Source data 1.** Raw flow cytometry plots over the long term experimental evolution.

*Figure 2 continued on next page*

*Figure 2 continued*

**Source data 2.** Population GFP Ridgeplots.

**Figure supplement 1.** Independent *GAP1* amplifications lacking CNV reporter amplification.

## Decreased CNV formation rates in modified genomes suggests adjacent genomic elements contribute to *GAP1* CNV formation

We used nnSBI to infer CNV formation rates and selection coefficients from the evolutionary dynamics observed in glutamine-limited chemostats (*Figure 1B*). Previously, nnSBI estimations have been experimentally validated demonstrating its accuracy and reliability (*Avecilla et al., 2022*). First, we trained a neural density estimator using evolutionary simulations (Methods). This neural density estimator then allows us to infer posterior distributions and estimate the model parameters (i.e. the *GAP1* CNV formation rate, $\delta_C$; the *GAP1* CNV selection coefficient, $s_C$) from a single population CNV dynamics. We also inferred a collective posterior distribution from a set of replicate populations of the same strain. This collective posterior distribution consolidates estimations of the formation rate and selection coefficient from multiple replicate populations of one strain into one single estimate per strain in order to compare between the four strains, rather than between all 27 populations. We evaluated the confidence of our inference approach on synthetic simulations by computing its coverage, that is the probability that the true parameter falls within the 95% highest density interval (HDI) of the posterior distribution, a measure of certainty in an estimate similar to confidence intervals (*Kruschke, 2021*; *Supplementary file 2*). We find that the posterior distributions are narrow as the 95% HDI are less than an order of magnitude for both $s_C$ and $\delta_C$. Thus, we did not apply post-training adjustments to the neural density estimator, such as calibration (*Cook et al., 2006*) or ensembles (*Caspi et al., 2023*; *Hermans et al., 2022*) when estimating $\delta_C$ and $s_C$ from experimental *GAP1* CNV dynamics.

We find that the individual maximum a *posteriori* (MAP) estimates vary across strains and replicates (*Figure 3—figure supplement 2*). Overall, the CNV selection coefficient, $s_C$, ranges from 0.1 to 0.22 (with one exception of 0.3), whereas the CNV formation rate, $\delta_C$, ranges from $10^{-6}$ to $10^{-4}$ (with one exception of $10^{-3}$ and two of $10^{-7}$); and the proportion of early-occurring *GAP1* CNVs without amplification of the reporter ($\varphi$ ranges from $10^{-6}$ to $10^{-2}$ (with two exceptions of $10^{-8}$). We found that MAP estimates of replicate populations of the same strain cluster together, with some outliers (*Figure 3—figure supplement 2*). We performed posterior predictive checks, drawing parameter values from the posterior distributions and simulating the CNV dynamics (*Figure 3—figure supplement 3*), which agree with the observed data (*Figure 3—figure supplement 1*). For each strain, we use all individual posterior distributions to infer the collective posterior distribution, which is a posterior distribution conditioned on all observations, $P\left(\theta|X_1,\ldots,X_n\right)$ (Methods). The collective posterior allows us to estimate whether there is a difference in CNV formation rate and fitness effect across the four strains.

Collective posterior HDIs are very narrow (*Figure 3A*), and samples are highly correlated, as expected for joint estimation of selection coefficients and beneficial mutation rates (*Gitschlag et al., 2023*). The collective MAP estimates of the CNV selection coefficient are similar for the WT and LTRΔ (0.182). For ARSΔ and ALLΔ, the selection coefficient is estimated to be lower, with values of 0.146 and 0.126, respectively. However, all four selection coefficients are still large, consistent with these populations containing *GAP1* CNVs that are highly beneficial under glutamine-limitation. The collective MAP estimate for the CNV formation rate in WT is $4.5 \cdot 10^{-5}$. By contrast, the CNV formation rate is markedly lower in all mutant strains ranging from $1 \cdot 10^{-5}$ for LTRΔ and ALLΔ to $2.4 \cdot 10^{-6}$ in ARSΔ. These results support our hypothesis that proximate sequence features facilitate *GAP1* CNV formation.

The collective MAP predictions reproduce the experimental observations. Other than the very final time point for the WT population, all collective MAP predictions lay within the interquartile ranges (*Figure 3—figure supplement 1*). The observed *GAP1* CNV proportion stabilizes at different levels in the different experiments (*Figure 1B*). This can be explained by pre-existing or early-occurring unreported CNVs with proportions estimated to be between $\varphi{=}4 \cdot 10^{-6}$ to $1.6 \cdot 10^{-4}$ by the collective MAPs (*Figure 3—figure supplement 4*). Indeed, our model predicts that the total (reported and unreported) final CNV proportion is nearly one in all cases (*Figure 3—figure supplement 5*).

We sought to understand the consequences of differences in CNV formation rate and CNV fitness effects between the four strains on evolutionary dynamics. We used a modified version of the

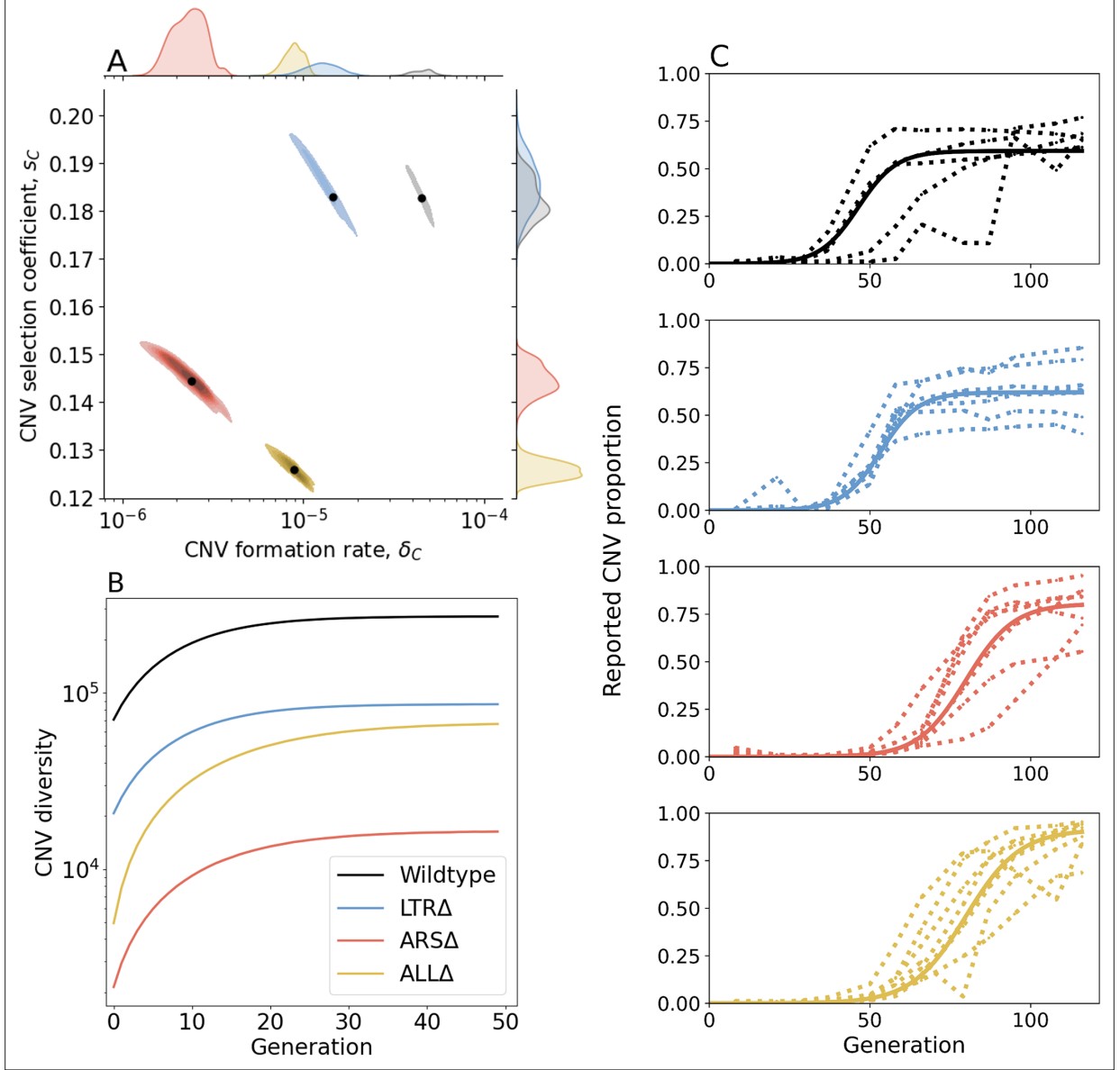

**Figure 3.** Inference of CNV formation rate and selection coefficient from experimental evolutionary data. (**A**) Collective MAP estimate (black markers) and 50% HDR (colored areas) of *GAP1* CNV formation rate, $\delta_C$, and selection coefficient, $s_C$. Marginal posterior distributions are shown on the top and right axes. (**B**) Collective posterior prediction of Shannon diversity of CNV lineages ($e^{-\Sigma_i[p_i log(p_i)]}$, *Jost, 2006*). Line and shaded area show mean and 50% HDI. (**C**) CNV reported frequency ($X_{C+}$) prediction using collective MAP (solid line) compared to empirical observations (dotted lines).

The online version of this article includes the following figure supplement(s) for figure 3:

**Figure supplement 1.** Posterior predictive checks for all replicates.

**Figure supplement 2.** MAP estimates of *GAP1* CNV formation rates ($\delta_C$) and selection coefficients ($s_C$) for all replicate populations.

**Figure supplement 3.** Error estimation of parameter inference.

**Figure supplement 4.** Pairwise and marginal collective posteriors for all estimated model parameters.

**Figure supplement 5.** Total *GAP1* CNV frequency.

**Figure supplement 6.** Pairwise evolutionary competition predictions.

**Figure supplement 7.** Neural density estimator training and validation loss during training.

**Figure supplement 8.** Parameter estimation accuracy on synthetic data.

evolutionary model with the estimated parameters to simulate an evolutionary competition between WT and the three architecture mutant strains over 116 generations, a point at which CNVs have reached high proportions in the experiment. To 'win' these competitions, the competitor strains need to adapt to glutamine-limitation by producing CNVs. The results of the simulated competitions predict that the WT outcompetes the other strains in all cases as its predicted final proportion almost always exceeds its initial proportion of 0.5 (*Figure 3—figure supplement 6* and *Supplementary file 3*). The average predicted proportion of WT cells when competing with LTRΔ is 0.717. By contrast, ARSΔ and ALLΔ are predicted to be almost eliminated by generation 116, as the average predicted WT proportion is 0.998 and 0.999, respectively. These simulated competitions further suggest that the ARS is a more important contributor to adaptive evolution mediated by *GAP1* CNVs.

Next, we estimated *de novo* CNV diversity in each strain. Previous work showed a diversity of CNV alleles formed under glutamine-limited selection including tandem duplications, segmental amplification, translocations, and whole chromosome amplification (*Lauer et al., 2018*), and that lineage richness decreases rapidly over the course of evolution due in part to competition and clonal interference (*Lauer et al., 2018*; *Levy et al., 2015*; *Nguyen Ba et al., 2019*). Our model does not include competition, clonal interference, or recurrent CNV formation. Therefore, diversity calculations are likely overestimations. Nonetheless, a comparison of diversity between strains is informative of whether proximate genome elements affect CNV allele diversity. Therefore, for each strain, we used its collective MAP to simulate a posterior prediction for the genotype frequencies (*Figure 3C*), which we then used to predict the posterior Shannon diversity (*Jost, 2006*). In all populations, we predict the set of CNV alleles to be highly diverse: the final predicted Shannon diversity ranges from $1.6 \cdot 10^4$ in ARSΔ to $3.2 \cdot 10^5$ in WT (*Figure 3B*). Our model predicts that the diversity increases rapidly during the selection phase and stabilizes in the equilibrium phase. This is because CNV alleles that form towards the end of the experiment would have a low frequency with a minor effect on diversity. We observe the greatest diversity in WT populations with lower diversity in the three genomic architecture mutants. Moreover, diversity saturates faster in WT populations. This suggests that the WT strain is able to form more unique CNVs allele types earlier compared to the other three strains (*Figure 3B*). Shannon diversity is lower in LTRΔ and further lower in ALLΔ and ARSΔ (*Figure 3B*) reflecting the rank order of CNV formation rates (*Figure 3A*).

## Inference of CNV mechanisms in genome architecture mutants

Contrary to our expectations, removal of proximate genomic elements from the *GAP1* locus does not inhibit the formation of *GAP1* CNVs. We sought to determine the molecular basis by which *GAP1* CNVs form in the absence of these local elements. Therefore, we isolated 177 *GAP1* CNV-containing clones across each population containing the four different strains at generations 79 and 125 and performed Illumina whole-genome sequencing. Using a combination of read depth, split read, and discordant read analysis, we defined the extent of the amplified region, the precise CNV breakpoints, and *GAP1* copy number. On the basis of these features, we inferred the CNV-forming mechanisms for each *GAP1* CNV (**Methods**). Among the 177 analyzed *GAP1* CNVs, we observed tandem amplifications, tandem triplications with an inverted middle copy, intra- and inter-chromosomal translocations, aneuploidy, and complex CNVs. *GAP1* copy numbers range from two to six in any given clone. Each of the four strains is able to produce a diversity of CNV alleles ranging from small (tens of kilobases) to large (~hundreds of kilobases) segmental amplifications (*Figure 4*). We quantified the CNV length per strain (*Figure 4E*) and found no significant relationship between CNV length and generation from which the clone was isolated (ANOVA, p=0.33) and therefore considered all 177 clones in subsequent comparisons (*Figure 4—figure supplement 1*). We found no significant effect of the inferred $s_C$ (ANOVA, p=0.673) or $\delta_C$ (ANOVA, p=0.277) on CNV length. We defined six major CNV-forming mechanisms across the four strains: ODIRA, LTR NAHR, NAHR, transposon-mediated, complex CNVs, and whole chromosome duplication (aneuploidy) and assigned each CNV allele to one mechanism using diagnostic features of each CNV (*Figure 4A–D* and Methods).

## ODIRA is a predominant mechanism of CNV formation

We inferred *GAP1* CNVs formed through ODIRA in all four genotypes at high frequencies: 22 out of 37 WT clones (59%), 42 out of 52 LTRΔ clones (81%), 11 out of 42 ARSΔ clones (26%), and 12 out of 46 ALLΔ clones (26%). Considering the set of all CNVs in all strains, ODIRA is the most common CNV

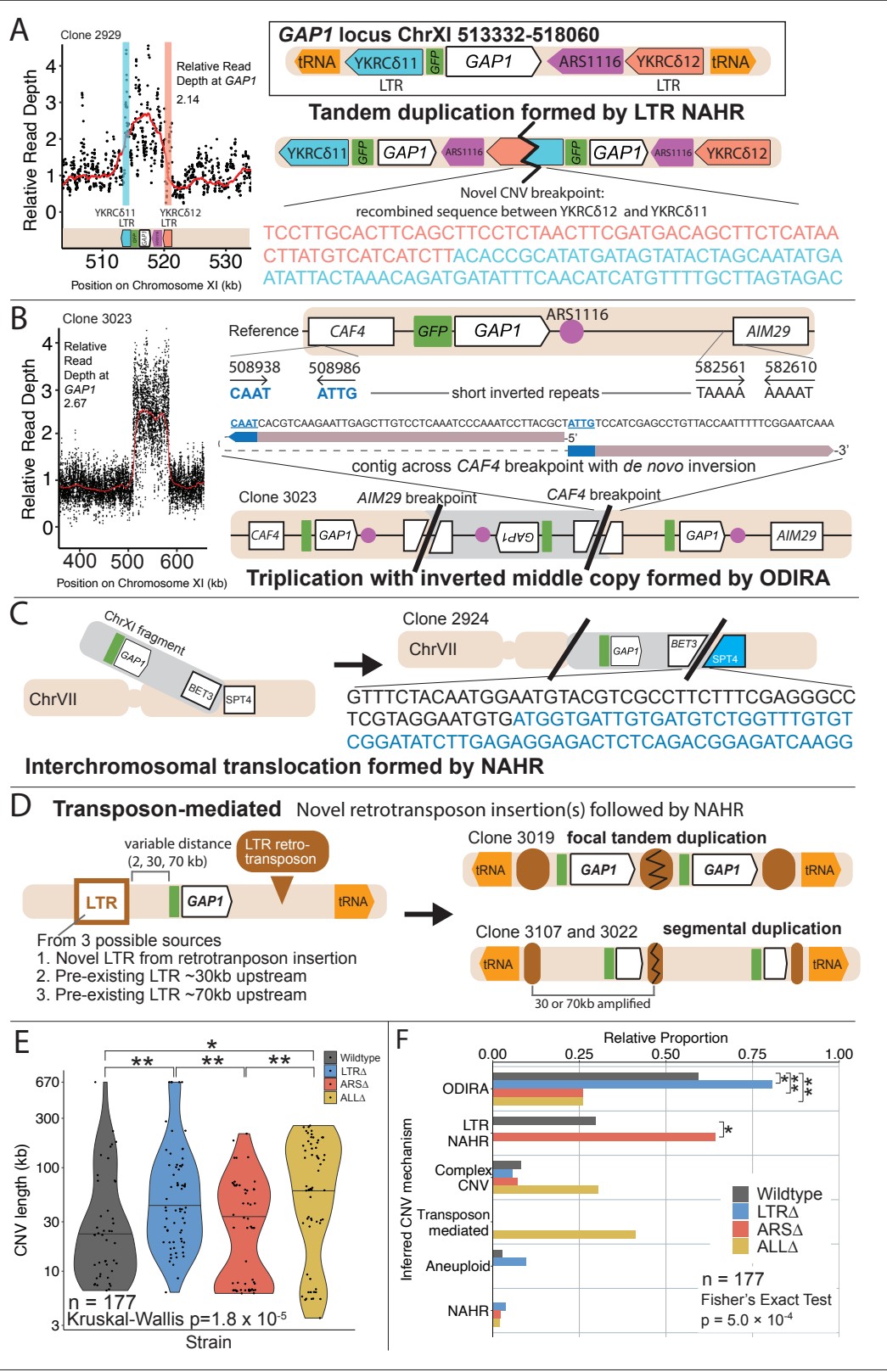

**Figure 4.** *GAP1* CNV alleles can be formed through a variety of mechanisms. (**A**) Schematic of *Saccharomyces cerevisiae GAP1* locus on Chromosome XI: 513332–518060 with LTR, ARS elements and tRNA genes labeled. Long terminal repeat non-allelic homologous recombination (LTR NAHR) is defined on the basis of both CNV breakpoints occuring at LTR sites as revealed by read depth plots (left, pink and blue vertical lines) and increased

*Figure 4 continued on next page*

*Figure 4 continued*

read depth relative to the genome-wide read depth (left). In some cases we detect a hybrid sequence between two LTR sequences, a result of recombination between the two LTRs (right). LTR NAHR typically forms tandem duplications. (**B**) ODIRA is a DNA replication-error based CNV mechanism generated by template switching of the leading and lagging strand template at short inverted repeats. In the clone 3023 example, the relative read depth estimate of 2.67 copies of *GAP1* is rounded to 3 copies (left) and has apparent breakpoints in the *CAF4* and *AIM29* genes. We classify a clone as being formed by ODIRA if it has a *de novo* inverted sequence in at least one breakpoint. In the clone 3023 example, the short inverted repeat pairs are CAAT, ATTG (ChrXI:508938, ChrXI: 508986) in *CAF4* and TAAAA, AAAAT (ChrXI:582561, ChrXI582610) in *AIM29*. The contig sequence at the breakpoint (rectangle) is aligned to a reference *CAF4* coding sequence fragment. The 5' and 3' ends of the contig are labeled and a dashed line indicates contiguity (no gaps). The contig spans the *CAF4* breakpoint junction and contains a *de novo* inversion, i.e. two fragments of the *CAF4* gene are in opposite orientations with the mediating short inverted repeats shown in blue and underlined. The contig was generated using CVish (Methods) and supported by split reads at the breakpoint junction (not shown, see JBrowse alignments in Data availability). The contig containing a *de novo* inversion across the *AIM29* breakpoint is not shown. ODIRA typically forms tandem triplications with an inverted middle copy and contains an ARS (bottom). (**C**) Non-allelic homologous recombination (NAHR) is defined by having at least one CNV breakpoint not at the proximate LTR sites, that is other homologous sequences in the genome. In the clone 2924 example, we detect a hybrid sequence between the two homologous sequences in *BET3* (ChrXI) and *SPT4* (ChrVI). Because these two sequences are on different chromosomes we infer that an interchromosomal translocation occurred. The other breakpoint is unresolved. A read depth plot supports an amplified segment containing the *GAP1* gene. NAHR is able to produce supernumerary chromosomes as is the case in Clone 2968 (*Figure 5A*). (**D**) Transposon-mediated mechanism is defined by an inference of at least one intermediate novel LTR retrotransposon insertion followed by LTR NAHR. In the ALLΔ strains which have the flanking LTRs deleted, we find novel LTR retrotransposon insertions near previously deleted LTR sites. The newly deposited LTR sequence (downstream of *GAP1*) (brown oval) recombines with another LTR sequence (upstream of *GAP1*). The upstream LTR (white and brown rectangle) is either pre-existing or introduced by a second *de novo* retrotransposition. Depending on the relative position of the upstream LTR to the *GAP1* gene, focal or segmental amplifications form. In the case of clone 3019, a focal amplification formed after two novel LTR transposition events flanking the *GAP1* gene and subsequent LTR NAHR. In the case of clones 3107 and 3022, segmental amplification formed after recombination between the downstream LTR and a pre-existing LTR sequence 30 and 70 kb upstream, respectively. Read depth estimation (not shown) supports CNV breakpoints at pre-existing or inferred newly deposited LTRs. (**E**) Violin plot of CNV length in each genome-sequenced clone, n=177. Strain has a significant effect on CNV length, Kruskal-Wallis test, p=1.008 x 10$^{-5}$. Pairwise wilcoxon rank sum test with bonferroni correction show significant CNV length differences between WT and LTRΔ (p=0.00490), WT and ALLΔ (p=0.01230), LTRΔ and ARSΔ (p=0.00056), ARSΔ and ALLΔ (p=0.002). (**F**) Barplot of inferred CNV mechanisms, described in A-D, for each CNV clone isolated from glutamine-limited evolving populations. Complex CNV is defined by a clone having more than two breakpoints in chromosome XI, indicative of having more than one amplification event. Inference came from a combination of read depth, split read, and discordant read analysis and the output of CVish (see Methods). Strain is significantly associated with CNV mechanism, Fisher's Exact Test, p=5.0 x 10$^{-4}$, n = 177. There is a significant increase in ODIRA prevalence between WT and LTRΔ, chi-sq, p=0.02469. There is a significant decrease in ODIRA prevalence from WT to ARSΔ and ALLΔ, chi-sq, p=0.002861 and 0.002196, respectively. There is a significant decrease of LTR NAHR from WT to LTRΔ, chi-sq, p=0.03083.

The online version of this article includes the following figure supplement(s) for figure 4:

**Figure supplement 1.** No significant interaction between strain and generation on CNV length.

**Figure supplement 2.** Types of ODIRA detected.

**Figure supplement 3.** CNV mechanisms in ARSΔ clones.

mechanism comprising almost half of all CNVs (87/177, 49%). The second most common mechanism occurs about half as often—NAHR between flanking LTRs (38/177, 21%), which generates tandem amplifications. In the WT background, ODIRA (22/37) and NAHR between LTRs (11/37) account for 89% of *GAP1* CNVs.

In LTRΔ populations, *GAP1* CNVs form via ODIRA, chromosome missegregation, and NAHR using other sites. As expected, in LTRΔ clones we did not detect NAHR between LTRs in 52 clones and no focal amplifications were detected (*Figure 5B*). In LTRΔ populations CNVs are formed predominantly by ODIRA (42/52, 81%) (*Supplementary file 4*), a significant increase relative to WT clones (chi-sq, p=0.02469; *Figure 4F*). By contrast, aneuploidy (5/52), complex CNV (3/52), and NAHR (2/52) account for less than 10% of *GAP1* CNVs in LTRΔ. Consequently, we observe an increase in average *GAP1* CNV

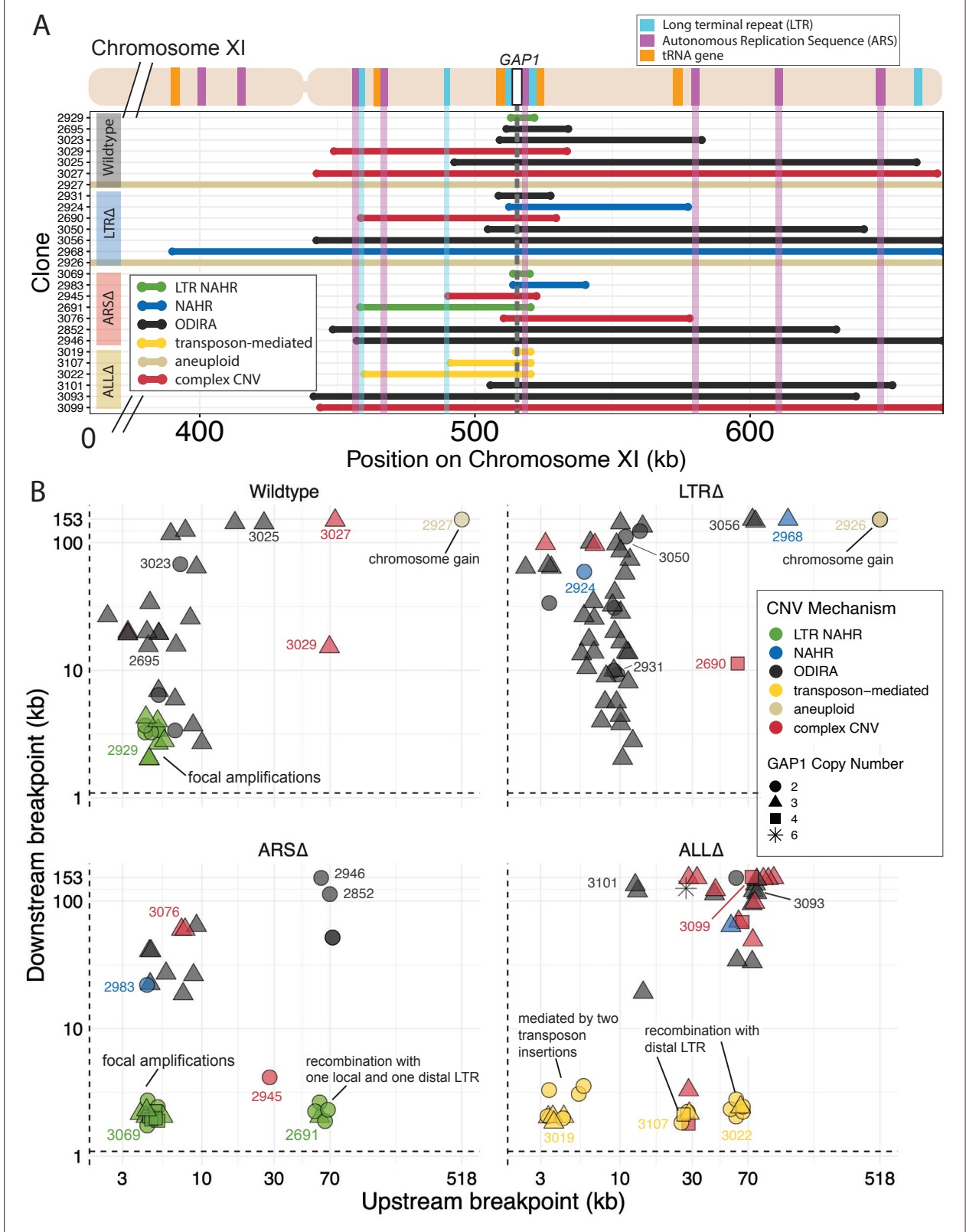

**Figure 5.** Local and distal elements contribute to the formation of *GAP1* CNV alleles. (**A**) Top: Schematic of *S. cerevisiae* chromosome XI, with LTR, ARS elements, tRNA genes annotated. LTR-blue, ARS-purple, tRNA-orange, *GAP1* ORF-white rectangle. Using a combination of read depth, split read, and discordant read analysis, we defined the extent of the amplified region, CNV breakpoints, and *GAP1* copy number. *GAP1* copy numbers were estimated using read depth relative to the average read depth of chromosome XI. Bottom: Dumbbell plots represent the amplified region (>1 copy) relative to the

*Figure 5 continued on next page*

*Figure 5 continued*

WT reference genome. The ends of the dumbbells mark the approximate CNV breakpoints shown relative to the start codon of the *GAP1* ORF (vertical dotted line). Select clones were chosen as representative of the observed diversity of amplifications in this study. (**B**) Scatterplots of CNV length for all genome-sequenced clones, n = 177. We defined the upstream and downstream breakpoints as kilobases away from the start codon of the *GAP1* ORF (vertical dashed line in **A** and this scatterplot). CNV mechanisms are defined in *Figure 4A-D* and Methods. Select clones from **A** are annotated with their clone ID number. Note the focal amplifications resulting from LTR NAHR in WT clones and ARSΔ clones, respectively. In ARSΔ, additionally note NAHR between one local and one distal LTR ~60 kb upstream. Note in ALLΔ, focal amplifications were likely mediated by two newly deposited LTR sequences from two transposon insertions. Additionally, note in ALLΔ amplifications formed between one newly inserted LTR and one distal pre-existing LTR sequence, ~30 kb or ~70 kb upstream.

length in LTRΔ relative to WT (*Figure 4E*) as there is an increased prevalence of segmental amplifications and aneuploidy (*Figure 5B*).

Overall, aneuploidy was observed infrequently. Whole amplification of chromosome XI was detected in six out of 177 clones (3.4%; *Figure 4F*) and detected in only two strains: WT and LTRΔ (*Figure 5A*). We also detected supernumerary chromosomes in five out of 177 clones (2.8%), which formed through both NAHR and ODIRA (*Figure 5A*).

## ODIRA generates CNVs using distal ARS

Whereas removal of proximate LTRs prevents the formation of small tandem duplication CNVs through LTR NAHR, removal of the local ARS does not prevent the formation of *GAP1* CNVs through ODIRA (*Figure 4F*). In the absence of the proximate ARS, distal ones are used to form ODIRA as all amplified regions of ODIRA clones contain a distal ARS (*Figure 5A*)**,** with one exception (Methods, *Figure 4—figure supplement 2*). We observe a significant increase of LTR NAHR in the ARSΔ clones (27/52, 52%) relative to WT clones (11/37, 39%) (*Figure 4F*, chi-sq, p=0.03083). In ARSΔ clones, we find two CNV length groups (*Figure 4E*) that correspond with two different CNV mechanisms (*Figure 4—figure supplement 3*). All smaller CNVs (6–8 kb; *Figure 4—figure supplement 3*) correspond with a mechanism of NAHR between LTRs flanking the *GAP1* gene (*Figure 5B*, ARSΔ**,** bottom left green points). Larger CNVs (8 kb-200kb; *Figure 4—figure supplement 3*) correspond with other mechanisms that tend to produce larger CNVs, including ODIRA and NAHR between one local and one distal LTR element (*Figure 5B*).

Surprisingly, we found CNVs with breakpoints consistent with ODIRA that contained only two copies of the amplified region, whereas ODIRA typically generates a triplication. In the absence of additional data, we cannot rule out inaccuracy in our read-depth estimates of copy numbers for these clones (ie. they have three copies). An alternate explanation is a secondary rearrangement of an original inverted triplication resulting in a duplication (*Brewer et al., 2024*); however, we did not detect evidence for secondary rearrangements in the sequencing data. Given that these clones have two inverted junctions consistent with ODIRA, we presume they must have three copies of the amplified region and the read depth data estimate is likely inaccurate for these clones. Notably, we found three additional ODIRA clones that end in native telomeres, each of which had amplified three copies. In these clones the other breakpoint contains the centromere, indicating the entire right arm of chromosome XI was amplified three times via ODIRA, each generating supernumerary chromosomes. Thus, ODIRA can result in amplifications of large genomics regions from segmental amplifications to supernumerary chromosomes.

## Novel retrotransposition events potentiate *GAP1* CNVs

CNVs in the ALLΔ clones form by two major mechanisms: (1) ODIRA using distal ARS sites to form large amplifications and (2) LTR NAHR following novel Ty LTR retrotransposon insertions to form focal amplifications (transposon-mediated, *Figure 4*). These two classes are evident in the broad CNV lengths detected (*Figure 4E*). ALLΔ clones tend to have more larger amplifications formed by ODIRA than ODIRA-generated amplifications in WT and LTRΔ (*Figure 5B*) because they encompass distal ARS and inverted repeats (*Figure 5A*). Surprisingly, we detected novel LTR retrotransposon events that generated new LTRs that subsequently formed *GAP1* CNVs through NAHR with a pre-existing LTR in the genome or an LTR from a second novel retrotransposition (*Figure 5B*). This explains the small focal amplifications detected in ALLΔ clones that are in some cases smaller than that of WT (*Figure 4E*). Regions upstream of tRNA genes are known to be hotspots for Ty retrotransposons (*Ji*

*et al., 1993*; *Mularoni et al., 2012*). We find the novel retrotransposons insert near one or both of the previously deleted LTR sites (*Supplementary file 5*), which flank *GAP1* and are downstream of tRNA genes (*Figure 1A*). We only detected novel retrotranspositions in ALLΔ populations. In total we detected 15 unique Ty retrotransposon insertion sites of which eight were upstream of the deleted LTR, YKRCδ11, and four were downstream of deleted LTR, YKRCδ12 (*Supplementary file 5*). The remaining two insertions were distal to the *GAP1* gene: one on the short arm and the second on the long arm of chromosome XI. Every novel insertion was upstream of an tRNA gene, consistent with the biased preference of Ty LTR insertions (*Ji et al., 1993*; *Mularoni et al., 2012*). Recombination between a new and preexisting LTR produces large amplifications whereas recombination between two newly inserted Ty1 flanking the *GAP1* gene forms focal amplifications of the *GAP1* gene (*Figure 5B*).

## Discussion

In this study, we sought to understand the molecular basis of repeated *de novo* amplifications and selection of the general amino acid permease gene, *GAP1*, in *S. cerevisiae* evolving under glutamine-limited selection. We hypothesized that a high formation rate of *GAP1* CNVs is due to the unique genomic architecture at the locus, which comprises two flanking long terminal repeats and a DNA replication origin. We used genetic engineering, experimental evolution, and neural network simulation-based inference to quantify *de novo* CNV dynamics and estimate the CNV formation rate and selection coefficient in engineered mutants lacking the proximate genome elements. We find that removal of these elements has a significant impact on *de novo* CNV dynamics, CNV formation rate, and selection coefficients. However, CNVs are formed and selected in the absence of these elements highlighting the plasticity of the genome and diversity of mechanisms that generate CNVs during adaptive evolution.

Despite their proximity to *GAP1* and previous studies demonstrating the prevalence of NAHR between repetitive sequences forming CNVs (*Dunham et al., 2002*; *Gresham et al., 2010*; *Todd et al., 2019*; *Zhang et al., 2013*), we found that flanking LTRs are not an essential driver of CNV formation. The *de novo* CNV dynamics of WT and LTRΔ populations are similar and we find that although the CNV formation rate is reduced, the effect is small. By contrast, a significantly decreased CNV formation rate and delayed CNV appearance time was observed in the absence of the ARS in ARSΔ and ALLΔ populations, which suggests that the local ARS is an important determinant of *GAP1* CNV-mediated adaptive dynamics. Furthermore, the significant delay in the time at which the CNV frequency reaches equilibrium in the ARSΔ compared to WT (*Figure 1E*) can be explained by both the estimated lower CNV formation rate and lower selection coefficient (*Figure 3A*). ODIRA was identified as the predominant CNV mechanism in sequence-characterized clones revealing that DNA replication errors, specifically template switching of the leading and lagging strands, are a common source of CNV formation during adaptive evolution.

Using nnSBI we inferred lower rates of CNV formation in all strains with modified genomes that may be informative of the rate at which specific mechanisms occur. The lower CNV formation rate in the LTRΔ strain could be a closer approximation of ODIRA formation rates at this locus as ODIRA CNVs are the predominant CNV mechanism in the LTRΔ strain (*Figure 4F*). Furthermore, the low formation rates in the LTRΔ relative to WT might suggest that the presence of the flanking long terminal repeats may increase the rate of ODIRA formation through an otherwise unknown combinatorial effect of DNA replication across these flanking LTRs and template switching at the *GAP1* locus. ARSΔ has the lowest CNV formation rate and it could be an approximation of the rates of NAHR between flanking LTRs and ODIRA at distal origins. We find that the ALLΔ has a higher CNV formation rate than the ARSΔ. One explanation for this is that the deletion of the flanking LTRs in ALLΔ gives opportunity for novel transposon insertions and subsequent CNV formation through LTR NAHR. Indeed, we find an enrichment of novel transposon-insertions in the ALLΔ (*Figure 4F*) and subsequent CNV formation through recombination of the Ty1-associated repeats (*Figure 5B*, **ALLΔ**). The sequential events of transposon insertion followed by LTR NAHR must occur at a high rate to explain the increased CNV rate in ALLΔ relative to ARSΔ. While remarkable, increased transposon activity is associated with nutrient stress (*Curcio and Garfinkel, 1999*; *Lesage and Todeschini, 2005*; *Todeschini et al., 2005*) and therefore this is a plausible explanation for the CNV rate estimated in ALLΔ. Additionally, ARSΔ clones rely more on LTR NAHR to form CNVs (*Figure 4F*). The prevalence of ODIRA in ARSΔ and ALLΔ are similar. LTR NAHR usually occurs after

double strand breaks at the long terminal repeats to give rise to CNVs (**Argueso et al., 2008**). Because we use haploid cells, such double strand break and homology-mediated repair would have to occur during S-phase after DNA replication with a sister chromatid repair template to form tandem duplications. Therefore the dependency on LTR NAHR to form CNVs and the spatial (breaks at LTR sequences) and temporal (S-phase) constraints could explain the lower formation rate in ARSΔ.

The genomic elements have clear effects on the evolutionary dynamics using simulated competitive fitness experiments. The similar selection coefficients in WT and LTRΔ suggest that CNV clones formed in these background strains are similar. Indeed, the predominant CNV mechanism in both is ODIRA followed by LTR NAHR (**Figure 4F**). Whereas LTR NAHR is abolished in the LTRΔ, it seems that CNVs formed by ODIRA allow adaptation to glutamine-limitation similar to WT. The lower selection coefficients in ARSΔ and ALLΔ suggest that *GAP1* CNVs formed in these strains have some cost. In a competition, they would get outcompeted by CNV alleles in the WT and LTRΔ background (**Figure 3—figure supplement 6**). Additionally, the local ARS, ARS1116, is a major origin (**McGuffee et al., 2013**) and ODIRA CNVs found around this origin corroborate its activity. The simulated competitions (**Figure 3—figure supplement 6**) further suggest that the ARS is a more important contributor to adaptive evolution mediated by *GAP1* CNVs.

The prevalence of ODIRA generated CNVs is a consequence of multiple DNA replication origins and pervasive inverted repeat sequences throughout the chromosome (**Figure 4**). In particular, breakpoint analysis of LTRΔ CNV clones show that ODIRA produces a continuum of CNV sizes along the short arm of chromosome XI. Downstream breakpoints of ODIRA-generated CNVs range from nearby the *GAP1* gene (~3 kilobases) to the right telomere of chromosome XI (153 kilobases; **Figure 5B**). The *S. cerevisiae* genome contains a high frequency of inverted repeats ranging from 3bp to 14bp throughout the genome (**Martin et al., 2024**), but longer repeats are more likely to be used in ODIRA (**Martin et al., 2024**). The ubiquity of inverted repeats is in stark contrast to the relative paucity of LTR sequences, which are dispersed throughout the genome. Thus, ODIRA supplies a diverse and high number of gene amplifications for selection to act on, setting the stage for genome evolution and adaptation. It appears that complex CNVs may include secondary rearrangements after an initial ODIRA event as most complex CNVs in this study include signatures of ODIRA events (**Supplementary file 6**). Further work needs to be done to resolve the CNV structure.

Consistent with previous reports of increased Ty insertions in *S. cerevisiae* under stress conditions (**Morillon et al., 2000**; **Morillon et al., 2002**), we observed novel retrotransposon insertions in populations evolved in glutamine-limited chemostats. Transposon insertions can be harmful and lead to loss-of-function mutations but are also a means of generating beneficial alleles including CNVs (**Blanc and Adams, 2003**; **Dunham et al., 2002**; **Gresham et al., 2008**; **Wilke and Adams, 1992**). We only detected novel Ty insertions in the ALLΔ strain. This is likely because regions upstream of tRNA genes are predisposed to transposition. Our detection of novel retrotransposon insertions is consistent with a previous experimental evolution study that suggested that Ty insertions were rare under constant nitrogen-limitation and substantially more common under fluctuating nitrogen limitation, in which cells experience total nitrogen starvation periodically (**Hays et al., 2023**). In that study, 898 novel Ty insertions were found across 345 clones (**Hays et al., 2023**) corresponding to an average of 2.6 insertions per genome This high insertion frequency is consistent with detecting novel insertions on either side of the *GAP1* gene that subsequently mediate a focal amplification via LTR NAHR. Importantly, the role of Ty differs in the two studies, as in our case beneficial CNV formed after novel retrotransposition through recombination of newly introduced repeat sequences, whereas **Hays et al., 2023** found Ty-associated null alleles that are beneficial in nitrogen-limited conditions. Together, these results reveal the different means by which retrotransposition can facilitate adaptive evolution.

Aneuploidy was not a major source of adaptation in our experiments as it was infrequently detected (n=6/177). This contrasts with studies suggesting aneuploidy is a rapid and transient route to adaptation over short evolutionary time scales (**Chen et al., 2012a**; **Chen et al., 2015**; **Chen et al., 2012b**; **Pavelka et al., 2010**; **Selmecki et al., 2006**; **Selmecki et al., 2015**; **Yona et al., 2012**). However, aneuploidy incurs a fitness cost (**Robinson et al., 2023**; **Tsai et al., 2019**; **Yang et al., 2021**) and therefore can be outcompeted by slow-forming but less costly beneficial mutations in large populations (**Kohanovski et al., 2024**). Our observed higher frequencies of focal and segmental amplifications may be because they are less costly than whole-chromosome amplifications.

A variety of DNA replication errors generate CNVs. Replication slippage at palindromic DNA and DNA repeats can cause fork stalling and downstream CNV formation (*Lee et al., 2007*; *Zhang et al., 2009b*). DNA repeats can form secondary structures like R loops, cruciforms, non-B DNA structures, and hairpins which stimulate CNV formation (*Gu et al., 2008*). Untimely replication, faulty fork progression, S-phase checkpoint dysfunction, defective nucleosome assembly, and DNA repeat sites including LTRs are sources of replication-associated genome instability (*Aguilera and García-Muse, 2013*).

Additional processes may also play a role. The *GAP1* gene is highly transcribed under glutamine-limitation (*Airoldi et al., 2016*) and transcription-replication collisions may fuel ODIRA CNV formation at this locus (*Lauer and Gresham, 2019*; *Wilson et al., 2015*). CNV formation can also be stimulated by transcription-associated replication stress and histone acetylation (*Hull et al., 2017*; *Salim et al., 2021*; *Whale et al., 2022*) and replication fork stalling at tRNA genes (*Osmundson et al., 2017*; *Yeung and Smith, 2020*). Testing the role of transcription in promoting the formation of adaptive CNVs warrants further investigation.

Recent work has proposed that ODIRA CNVs are a major mechanism of CNVs in human genomes (*Brewer et al., 2011*; *Brewer et al., 2015*; *Martin et al., 2024*). Studies of human and yeast genomes have typically considered homologous recombination as the predominant mechanism of CNV formation (*Lupski and Stankiewicz, 2005*). CNV hotspots identified in the human (*Chance et al., 1994*; *Lupski, 1998*; *Lupski and Stankiewicz, 2005*; *Pentao et al., 1992*) and yeast genomes are indeed mediated by NAHR of long repeat sequences (*Green et al., 2010*; *Gresham et al., 2010*). However, a focus on recombination-based mechanisms as a means of generating copy number variation may be the result of ascertainment bias or the comparative ease of studying the effect of long repeat sequences over short palindromic ones. Our study demonstrates that experimental evolution in yeast is a useful approach to elucidating the molecular mechanisms by which DNA replication errors generate CNVs.

## Methods

### Strains and media

All strains used in this study are provided in *Supplementary file 7*. Each of the three architecture mutants were constructed independently starting with the *GAP1* CNV reporter strain (DGY1657). The CNV reporter is 3.1 kb and located 1117 nucleotides upstream of the *GAP1* coding sequence. It consists of, in the following order, an *ACT1* promoter, mCitrine (GFP) coding sequence, *ADH1* terminator, and kanamycin cassette under control of a *TEF* promoter and terminator. To construct each deletion strain, we performed two rounds of transformations both using PCR amplified donor templates designed for homology-directed repair. The first transformation used a repair template containing a nourseothricin resistance cassette to replace the pre-existing kanamycin resistance cassette and *GAP1* gene. The repair template was designed to also delete the elements of interest (ie. ARS1116, both flanking LTRs (YKRCδ11, YKRCδ12), or both LTRs and ARS1116). The second transformation replaced the nourseothricin cassette with a kanamycin resistance cassette and *GAP1* gene thus yielding a genomic architecture Δ strain that is kanamycin(+) and nourseothricin(-). We confirmed scarless deletions with sanger sequencing and whole-genome-sequencing. Final identifiers are DGY1657 for the WT strain, DGY2076 for the LTRΔ strain, DGY2150 for the ARSΔ strain, and DGY2071 for the ALLΔ strain.

The zero-, one-, and two-copy GFP controls, DGY1, DGY500, and DGY1315, respectively, are described in *Lauer et al., 2018* and *Spealman et al., 2023* (*Lauer et al., 2018*; *Spealman et al., 2023*). Briefly, GFP under the *ACT1* promoter was inserted at neutral loci that do not undergo amplification in glutamine-limited continuous culture. 400 µM glutamine-limited media is described in *Lauer et al., 2018*.

### Long-term experimental evolution

We performed experimental evolution of 30 *S. cerevisiae* populations in miniature chemostats (ministats) for ~137 generations under nitrogen limitation with 400 µM glutamine as in *Lauer et al., 2018*. Of the 30 populations, there were three controls: one control population with no fluorescent reporter (DGY1), one with one GFP fluorescent reporter (DGY500), one with two GFP fluorescent reporters (DGY1315). The remaining 27 populations have the *GAP1* CNV GFP reporter. Of these, five

populations are WT (DGY1657), seven are LTRΔ (DGY2076), seven are ARSΔ (DGY2150), and eight are ALLΔ (DGY2071). We inoculated each ministat containing 20 ml of glutamine-limited media with 0.5 ml culture from its corresponding genotype founder population. The founder population was founded by a single colony grown overnight in glutamine-limited media at 30 °C. Replicate populations of the same strain were inoculated from the same founder population derived from a single colony. Strains were randomized among the 30-plex ministat setup to account for the possibility of systematic position effects. After inoculation, populations were incubated in a growth chamber at 30 °C for 24 hr with the media inflow pump off. After 24 hr, the populations had reached early stationary phase and we turned on the media inflow pump and waited 4 hr for the populations to reach steady-state equilibrium, at which the population size was ~$10^8$ cells. This was generation zero. Ministats were incubated in a growth chamber at 30 °C with a dilution rate of 0.12 culture volumes/hr. Since the ministats had a 20 ml culture volume, the population doubling time was 5.8 hr. Approximately every 10 generations, we froze 2 ml samples of each population in 15% glycerol stored at –80 °C. Approximately every 30 generations, we pelleted cells from 1 ml samples of each population and froze them at –80 °C for genomic DNA extraction.

## Flow cytometry analysis to study *GAP1* CNV dynamics

To track *GAP1* CNV dynamics, we sampled 1 ml from each population approximately every 10 generations. We sonicated cell populations for 1 min to remove any cell clumping and immediately analyzed samples on the Cytek Aurora flow cytometer. We sampled 100,000 cells per population and recorded forward scatter, side scatter, and GFP fluorescent signals for every cell. We performed hierarchical gating to define cells, single cells, unstained (zero-copy-GFP control) cells, cells with one copy of GFP (*GAP1*), and two or more copies of GFP (*GAP1*; *Spealman et al., 2023*). First we gated for cells (filtered out any debris, bacteria) by graphing forward scatter area (FSC-A) against side scatter area (SSC-A). Second, we gated for single cells by graphing forward scatter area against forward scatter height and drawing along the resulting diagonal. Finally, we drew non-overlapping gates to define three subpopulations: zero copy, one copy, and two or more copies of GFP by graphing B2 channel area (B2-A), which detects GFP (excitation = 516 $\lambda$, emission = 529 $\lambda$), against forward scatter area (FSC-A). We note that the one copy and two copy events overlap some, which is a limitation in this experiment (*Spealman et al., 2023*).

We found that two architecture mutants, DGY2150 and DGY2071, had strain-specific GFP fluorescence even though they only harbored one copy of GFP. DGY2150 and DGY2071 had slightly higher fluorescence than the one copy GFP control strain, DGY500, but less than that of the two copy GFP control strain, DGY1315. The third architecture mutant, DGY2076, had the same GFP fluorescence as the one-copy GFP control strain (DGY500). We ruled out that they were spontaneous diploids by looking at forward scatter signals. The forward scatter signal was not different from that of the one copy control (a haploid) and was not as high as a diploid. Therefore due to strain-specific fluorescence, we decided to perform strain-based gating, ie. one set of gates for the WT strain, a second set of gates for the LTRΔ strain, and so on. Since the controls are also a strain of their own, they were not used to set universal gates for one-copy or two-copy. Thus, for each strain, we chose the basis of our one-copy gate as the timepoint per strain in aggregate with the lowest median cell-sized normalized fluorescence. The two-or-more-copy (CNV) gate was drawn directly above and non-overlapping with the one-copy gate.

## Quantification of dynamics

To obtain the proportion of CNVs for each population at each timepoint, we applied gates that correspond to zero-, one-, and two-or-more copy subpopulations. Using such proportion per population per timepoint, we summarize population CNV dynamics as follows *Lauer et al., 2018*; *Spealman et al., 2023*. We calculate the generation of CNV appearance for each of the evolved populations. We defined CNV appearance as the generation where the proportion of CNV-containing cells first surpasses a threshold of 10% for three consecutive generations. Next, modified from *Lang et al., 2011* and *Lauer et al., 2018*, we calculate the percent increase in CNVs per generation for each evolved population. We compute the natural log of the proportion of the population with CNVs divided by the proportion of the population without CNVs for each timepoint. These proportions were obtained previously by gating. We plot these values across time and perform linear regression during

**Table 1.** Model parameters and priors.

Fixed parameters from *Avecilla et al., 2022*; *Hall et al., 2008*; *Joseph and Hall, 2004*; *Venkataram et al., 2016*.

| Parameter | Description | Prior / Fixed value |
|---|---|---|
| $s_C$ | GAP1 CNV selection coefficient | $log_{10}(s_C) \sim U[-2, 0]$ |
| $\delta_C$ | GAP1 CNV formation rate | $log_{10}(\delta_C) \sim U[-7, -0.3]$ |
| $\varphi$ | Proportion of pre-existing cells with GAP1 CNV | $log_{10}(\varphi) \sim U[-8, -2]$ |
| $s_B$ | Beneficial SNV selection coefficient | $10^{-3}$ |
| $\delta_B$ | Beneficial SNV formation rate | $10^{-5}$ |

the initial increase of CNVs. The slope of the linear regression is the percent increase in CNVs per generation. Finally, we calculate the time to CNV equilibrium, as defined by the generation at which a linear regression results in a slope <0.005 after the selection phase.

## Neural network simulation-based inference of evolutionary parameters

### Evolutionary model

We developed a Wright-Fisher model that describes the evolutionary dynamics, similar to our previous study (*Avecilla et al., 2022*). In that study we have shown that a Wright-Fisher model is suitable for describing evolutionary dynamics in a chemostat. Wright Fisher is a discrete-time evolutionary model with a constant population size and non-overlapping generations. Every generation has three stages: selection, in which the proportion of genotypes with beneficial alleles increases; mutation, in which genotypes can gain a single beneficial mutation or CNV; and drift, in which the population of the next generation is generated by sampling from a multinomial distribution. Our model follows the change in proportion of four genotypes (*Figure 2B*): $A$, the ancestor genotype; $B$, a cell with a non-CNV beneficial mutation; $C^+$, a genotype with two copies of *GAP1* and two copies of the CNV reporter; and $C^-$, a genotype with two copies of *GAP1* but only a single copy of the CNV reporter. CNV and non-CNV alleles are formed at a rate of $\delta_C$ and $\delta_B$ and have a selection coefficient of $s_C$ and $s_B$, respectively. The proportion of genotype $i$ is $X_i$. Unlike $X_B$ and $X_{C^+}$, which may increase due to both mutation and selection, we assume that $C^-$ is not generated after generation 0 (as experimental results suggest that the reporter is working properly). Hence, the proportion of the $C^-$ genotype only increases due to selection, with $s_C$ as its selection coefficient. We assume $C^-$ has an initial proportion $\varphi$. Model equations and further details are in the *Supplementary file 3*.

### Simulation-based inference

We use a neural network simulation-based inference method, Neural Posterior Estimation or NPE (*Papamakarios, 2019*) to estimate the joint posterior distribution of three model parameters, $s_C$, $\delta_C$ and $\varphi$, while the other parameters, $s_B$ and $\delta_B$ are fixed to a specific value (*Table 1*). Inferring all five model parameters resulted in similar prediction accuracy and $s_C$ and $\delta_C$ estimates.

We applied NPE, implemented in the Python package *sbi* (*Tejero-Cantero et al., 2020*), using a masked autoregressive flow (*Papamakarios et al., 2018*) as the neural density estimator: an artificial neural network that 'learns' an amortized posterior of model parameters from a set of synthetic simulations. Posterior amortization allows us to infer the posterior distribution $P(\theta|X)$ for a new observation $X$ without the need to re-run the entire inference pipeline, that is generating new simulations and re-training the network (as is the case in sampling-based methods such as Markov chain Monte Carlo or MCMC).

We generated 100,000 synthetic observations simulated from our evolutionary model using parameters drawn from the prior distribution (*Table 1*). The neural density estimator was trained using early stopping with a convergence threshold of 100 epochs without decreases in minimal validation loss (the default in *sbi* is 20). Using 100 epochs as a threshold resulted in improved predictions. We validated that this improvement in prediction accuracy is not a result of over-fitting (*Figure 3—figure supplement 7*).

We validated the trained neural density estimator by measuring the coverage property: the probability that parameters fall within the inferred posterior marginal 95% HDI. Then, we used the distribution of $\left(\frac{MAP}{True}\right)$ (**Figure 3—figure supplement 8**) and posterior predictive checks (**Figure 3—figure supplement 1**) as quantitative and qualitative measures of prediction accuracy, respectively.

## Collective posterior distribution

NPE estimates a single posterior distribution per observation, that is $P\left(\theta|X\right)$. Given $n$ observations $X_1, \ldots X_n$ generated from the same model distribution $P\left(\theta\right) P\left(X|\theta\right)$, where each observation is a time-series of GAP1 CNV proportion, NPE infers $n$ individual posterior distributions, each conditioned on a single observation, $P\left(\theta|X_i\right)$. We infer the *collective posterior distribution* based on $n$ individual posteriors, that is, a posterior distribution conditioned on all observations,

$$P\left(X_1, \ldots, X_n\right) = \frac{P\left(\theta\right)^{1-n} \Pi_i \left[P\left(\theta|X_i\right)\right]}{\int P\left(\zeta\right)^{1-n} \Pi_i \left[P\left(\zeta|X_i\right)\right] \, d\zeta} \tag{1}$$

This can be computed using the individual posteriors $P\left(\theta|X_i\right)$ and the prior $P\left(\theta\right)$ (see **Supplementary file 3** for derivation). However, as $P\left(\theta|X_i\right)$ could be infinitesimally small, a single observation could potentially reject a parameter value that is likely according to other observations. We want the collective posterior to be robust to such non-representative observations. Therefore, we define $P_\epsilon\left(\theta|X_i\right) = max\left(\epsilon, P\left(\theta|X_i\right)\right)$ and use this quantity instead of $P\left(\theta|X_i\right)$ in **Equation 1**. For a correct choice of $\epsilon$, the collective posterior mode should reflect a value with high posterior density for multiple observations, rather than a value that no individual posterior completely rejects. We set $\epsilon = e^{-150}$ based on a visual grid-search. To find the normalizing factor (denominator in **Equation 1**), the integral is approximated by a dense Riemann sum ($300^3$ points). Maximizing the distribution, that is finding the collective MAP, is implemented using *scipy*'s *minimize* method with the Nelder-Mead algorithm.

## Genetic diversity

Using our evolutionary model with the inferred parameters, we can estimate the diversity of CNV alleles in the experiments. For each strain, we used samples from its collective posterior to simulate a posterior prediction for the CNV allele frequencies (**Figure 3C**), which we then used to compute the posterior Shannon diversity (**Jost, 2006**), as detailed in the **Supplementary file 3**.

## Whole genome sequencing of isolated clones

Clones were isolated from archived populations and verified to harbor a *GAP1* CNV by measuring GFP fluorescence signal consistent with two or more copies. Populations of each strain from generation 79 were streaked out from the –80 °C archive on YPD and incubated at 30 °C for 2 days. Plates containing single colonies were viewed under a blue light to view GFP fluorescent colonies by eye. Relative to the fluorescence of the 2 copy control strain, we picked single colonies that fluoresced as bright or brighter, reasoning that these colonies would likely contain *GAP1 CNVs*. Single colonies were used to inoculate cultures in glutamine-limited media and incubated at 30 °C for 18 hr. The cultures were analyzed on the Cytek Aurora to verify they indeed harbored two or more copies of *GAP1* based on GFP fluorescence signal. For Illumina whole genome sequencing, genomic DNA was isolated using Hoffman-Winston method. Libraries were prepared using a Nextera kit and Illumina adapters. Libraries were sequenced on Illumina NextSeq 500 platform PE150 (2x150 300 Cycle v2.5) or Illumina NovaSeq 6000 SP PE150 (2x150 300 Cycle v1.5). We also used custom Nextera Index Primers reported in S1 **Baym et al., 2015** (**Baym et al., 2015**).

## Breakpoint analysis and CNV mechanism inference in sequenced clones

### Reference genomes

We created a custom reference genome for each of the genomic architecture mutants. The custom reference genome containing the *GAP1* CNV reporter in **Lauer et al., 2018** (NCBI assembly R64) was modified to delete the flanking LTRs, single ARS, or all three elements.

## Copy number estimation by read depth

The estimation of *GAP1* copy number from read depth used is described in *Lauer et al., 2018*, except we searched for ≥1000 base pairs of contiguous sequence. CNV boundaries were refined by visual inspection.

## Structural variation calling and breakpoint analysis

Whole genome sequences of clones were run through CVish version 1.0, a structural variant caller (*Spealman, 2019*). Structural variant calling was also done on each of the ancestor genomes: WT, ARSΔ, LTRΔ, and ALLΔ. Output.bam files containing split reads and discordant reads of evolved clones and their corresponding ancestor were visualized on Jbrowse2 or IGV to confirm locations of *de novo* CNV breakpoints and orientation of sequences at the novel breakpoint junctions. Novel contigs relative to the reference genome were outputted in addition to the supporting split reads that generated the contig. Blastn was used to verify orientation of contigs, namely inverted sequences used to define ODIRA (see Definitions of Inferred CNV Mechanisms). .bam files for each analyzed evolved clone and ancestors are available for view (See Data Availability).

## Definitions of inferred CNV mechanisms

We used the following liberal classifications for each CNV category. We called a clone ODIRA if we found inverted sequences in at least one breakpoint (*Figure 4A*). We define LTR NAHR as having both breakpoints at LTR sites (*Figure 4A*), evidence of recombination between the homologous LTR sequences. This mechanism typically forms tandem amplifications. In some cases, we find the hybrid sequence between two LTRs, but this is hard to recover in short-read-sequencing. We define NAHR as having breakpoints at homologous sequences, with at least one breakpoint not at an LTR sequence (*Figure 4A*). We define transposon-mediated as a clone having a breakpoint at a novel LTR retrotransposon site and the other breakpoint at a different LTR site (*Figure 4A*). Such characteristics support that the newly deposited LTR sequence recombined with another LTR sequence (either pre-existing or introduced by a second *de novo* retrotransposition) to form CNVs. Rarely, we are able to recover the hybrid sequence between LTR sequences even with high sequencing coverage 80–100 X. We define complex CNV as having more than two breakpoints on chromosome XI and a read depth profile that suggests more than one amplification event occurred (i.e. multi-step profile). For the complex CNV clones, we were not able to resolve the CNV mechanisms due to the limitations of short-read sequencing, though most have at least one ODIRA breakpoint.

## Acknowledgements

We thank all the members of the Gresham lab and Federica Sartori for helpful discussions, NYU Gencore for sequencing samples, and NYU High Performance Cluster for computing and storage. We thank Joshua Caleb Macdonald, Saharon Rosset, Uri Obolski, and Adi Stern for discussions and advice. This work was supported by NSF GRFP DGE1839302 (JNC), NIGMS T32GM132037 (JNC), NIGMS R01GM134066 (DG), R01GM107466 (DG), and R35GM153419 (DG), NIAID R01AI140766 (DG), NSF 1818234 (DG), Israel Science Foundation (ISF, YR 552/19), US–Israel Binational Science Foundation (YR & DG 2021276), Minerva Center for Live Emulation of Evolution in the Lab (YR) fellowship from the Edmond J Safra Center for Bioinformatics at Tel-Aviv University (NBN), and fellowship from the AI and Data Science Center at Tel-Aviv University (NBN).

# Additional information

### Funding

| Funder | Grant reference number | Author |
|---|---|---|
| National Science Foundation | DGE1839302 | Julie N Chuong |
| National Institute of General Medical Sciences | R35GM153419 | David Gresham |

| Funder | Grant reference number | Author |
|---|---|---|
| National Institute of General Medical Sciences | T32GM132037 | Julie N Chuong |
| National Institute of General Medical Sciences | R01GM134066 | David Gresham |
| National Institute of General Medical Sciences | R01GM107466 | David Gresham |
| National Institute of Allergy and Infectious Diseases | R01AI140766 | David Gresham |
| National Science Foundation | 1818234 | David Gresham |
| Israel Science Foundation | 552/19 | Yoav Ram |
| US–Israel Binational Science Foundation | 2021276 | Yoav Ram David Gresham |
| Minerva Stiftung | Minerva Center for Live Emulation of Evolution in the Lab | Yoav Ram |

The funders had no role in study design, data collection and interpretation, or the decision to submit the work for publication.

## Author contributions

Julie N Chuong, Conceptualization, Resources, Data curation, Software, Formal analysis, Supervision, Funding acquisition, Validation, Investigation, Visualization, Methodology, Writing – original draft, Project administration, Writing – review and editing; Nadav Ben Nun, Formal analysis; Ina Suresh, Julia Cano Matthews, Titir De, Grace Avecilla, Farah Abdul-Rahman, Nathan Brandt, Investigation; Yoav Ram, Software, Formal analysis, Supervision, Funding acquisition, Writing – review and editing; David Gresham, Conceptualization, Supervision, Funding acquisition, Investigation, Methodology, Project administration, Writing – review and editing

## Author ORCIDs

Julie N Chuong ⬥ http://orcid.org/0000-0002-4388-9458
Nadav Ben Nun ⬥ http://orcid.org/0009-0003-3228-7720
Yoav Ram ⬥ http://orcid.org/0000-0002-9653-4458
David Gresham ⬥ https://orcid.org/0000-0002-4028-0364

Reviewer #1 (Public review): https://doi.org/10.7554/eLife.98934.3.sa1
Reviewer #2 (Public review): https://doi.org/10.7554/eLife.98934.3.sa2
Reviewer #4 (Public review): https://doi.org/10.7554/eLife.98934.3.sa3
Author response https://doi.org/10.7554/eLife.98934.3.sa4

# Additional files

## Supplementary files

Supplementary file 1. Summary of genome sequence analysis of clones containing a single copy of the *GAP1* CNV reporter. Estimated copy number of the *GAP1* gene and inserted GFP gene of sequenced clones from five 1-copy-GFP minor subpopulations of the WT genome architecture strain. Copy number estimation is defined as the read depth of the target gene relative to the average read depth of the chromosome XI. Populations 1, 2, 4, 5 contain clones harboring GAP1 *CNVs* but only 1 copy of GFP. Clones from population 3 and 5 harbor 1 copy each of *GAP1* and GFP suggesting these lineages have beneficial mutations elsewhere in the genome, allowing coexistence with the *GAP1* CNV major subpopulation. CN, copy number, CNV = copy number variant, *GAP1*=general amino acid permease gene.

Supplementary file 2. Estimation of network confidence. The coverage, defining the probability that the true parameter falls within the 95% highest density interval (HDI) of the posterior distribution, for 829 synthetic simulations in which the final reported *GAP1* CNV proportion is at least 0.3. 95%

HDI was calculated for each simulation using 200 posterior samples. Our neural density estimator is slightly over-confident for $\varphi$ (coverage of 0.934), and under-confident for *GAP1* CNV selection coefficient and formation rate (coverage of 0.992 for $s_C$ and 0.995 for $\delta_C$). Despite this under-confidence, the posterior distributions are narrow in biological terms: the 95% HDI represents less than an order of magnitude for both $s_C$ and $\delta_C$. Thus, we did not apply post-training adjustments to the neural density estimator, such as calibration (*Cook et al., 2006*) or ensembles (*Caspi et al., 2023*; *Hermans et al., 2022*).

Supplementary file 3. Information and equations for the evolutionary model, simulated competitions, collective posterior distribution, and genetic diversity estimations.

Supplementary file 4. Inferred CNV mechanisms by strain. Counts of inferred CNV mechanisms for each sequenced clone, n=177, separated by strain

Supplementary file 5. Ty-associated clones and locations of novel Ty insertions.

Supplementary file 6. CNV Clone Sequencing Analysis.

Supplementary file 7. Strains used in this study.

MDAR checklist

## Data availability

Sequencing data is available at SRA PRJNA1098800.Other associated data are available here: https://osf.io/js7z8/. Source code repository simulation-based inference: https://github.com/yoavram-lab/chuong_et_al (copy archived at *Ben Nun, 2024*). Scripts for flow cytometry-based evolutionary dynamics and analysis of CNV clones: https://github.com/GreshamLab/local_arch_variants (copy archived at *Chuong, 2024*). Whole genome, split, discordants read depth profiles in the form of .bam files for each CNV strain and their corresponding ancestor aligned to our custom GFP GAP1 reference strain are displayed on https://jbrowse.bio.nyu.edu/gresham/?data=data/ee_gap1_arch_muts for WT strains, https://jbrowse.bio.nyu.edu/gresham/LTRKO_clones for LTRΔ strains, https://jbrowse.bio.nyu.edu/gresham/ARSKO_clones for ARSΔ strains, https://jbrowse.bio.nyu.edu/gresham/ALLKO_clones for ALLΔ strains.CNV breakpoints and associated information for all 177 clones are available in *Supplementary file 2*.

The following datasets were generated:

| Author(s) | Year | Dataset title | Dataset URL | Database and Identifier |
|---|---|---|---|---|
| Chuong J, Gresham D | 2024 | Local genome architecture on copy number variant dynamics | https://www.ncbi.nlm.nih.gov/bioproject/?term=PRJNA1098800 | NCBI BioProject, PRJNA1098800 |
| Chuong J | 2024 | DNA replication errors are a major source of adaptive gene amplification | https://osf.io/js7z8/ | Open Science Framework, js7z8 |

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
