## [Editor Report · eLife Assessment]

This study provides **important** new insights into the contributions of local DNA features to the complex molecular mechanisms and dynamics of copy number variation (CNV) formation during adaptive evolution. While limited to a single CNV of interest, the study is well-designed and carefully controlled, presenting **compelling** evidence that supports the conclusions. This work will be of general interest to those studying genome architecture and evolution from yeast biologists to cancer researchers.

---

## [Referee Report · Reviewer #1 (Public review)]

Summary:

The work by Chuong et al. provides important new insights into the contribution of different molecular mechanisms in the dynamics of CNV formation. It will be of interest to anyone curious about genome architecture and evolution from yeast biologists to cancer researchers studying genome rearrangements.

Strengths:

Their results are especially striking in that the "simplest" mechanism of GAP1 amplification (non-allelic homologous recombination between the flanking Ty-LTR elements) is not the most common route taken by the cells, emphasizing the importance of experimentally testing what might seem on the surface to be obvious outcome. One of the important developments of their work is the use of their neural network simulation-based inference (nnSBI) model to derive rates of amplicon formation and their fitness effects.

Weaknesses:

The nnSBI model that derives rates of amplicon formation and fitness is still opaque to this reviewer. All of the other criticisms made in the first review have been clarified/corrected in this much-improved version of the manuscript.

---

## [Referee Report · Reviewer #2 (Public review)]

Summary:

This study examines how local DNA features around the amino acid permease gene GAP1 influence adaptation to glutamine limited conditions through changes in GAP1 Copy Number Variation (CNV). The study is well motivated by the observation of numerous CNVs documented in many organisms, but difficulty in distinguishing the mechanisms by which they are formed, and whether or how local genomic elements influence their formation. The main finding is convincing and is that a nearby Autonomous Replicating Sequence (ARS) influences the formation of GAP1 CNVs and this is consistent with a predominate mechanism of Origin Dependent Inverted Repeat Amplification (ODIRA). These results along with finding and characterizing other mechanisms of GAP1 CNV formation will be of general interest to those studying CNVs in natural systems, experimental evolution and in tumor evolution. While the results are limited to a single CNV of interest (GAP1), the carefully controlled experimental design and quantification of CNV formation will provide a useful guide to studying other CNVs and CNVs in other organisms.

Strengths:

The study was designed to examine the effects of two flanking genomic features next to GAP1 on CNV formation and adaptation during experimental evolution. This was accomplished by removing two Long Terminal Repeats (LTRs), removing a downstream ARS, and removing both LTRs and the ARS. Although there was some heterogeneity among replicates, later shown to include the size and breakpoints of the CNV and the presence of an unmarked CNV, both marker assisted tracking of CNV formation and modeling of CNV rate and fitness effects showed that deletion of the ARS caused a clear difference compared to the control and the LTR deletion.

The consequence of deletion of local features (LTR and ARS) was quantified by genome sequencing of adaptive clones to identify the CNV size, copy number and infer the mechanism of CNV formation. This greatly added value to the study as it showed that (i) ODIRA was the most common mechanism but ODIRA is enhanced by a local ARS, (ii) non-allelic homologous recombination (NAHR) is also used but depends on LTRs, and (iii) *de novo* insertion of transposable elements mediate NAHR in strains with both ARS and LTR deletions. Together, these results show how local features influence the mechanism of CNV formation, but also how alternative mechanism can substitute when primary ones are unavailable.

Weaknesses:

The CNV mutation rate and effect on fitness are hard to disentangle. The frequency of the amplified GFP provides information about mutation rate differences as well as fitness differences. The data and analysis show that each evolved population has multiple GAP1 CNV lineages within it, with some being unmarked by GFP. Thus, estimates of CNV fitness are more of a composite view of all CNV amplifications increasing in frequency during adaptation. Another unknown but potential complication is whether the local (ARS, LTR) deletions influence GAP1 expression and thus the fitness gain of GAP1 CNVs. The neural network simulation based inference does a good job at estimating both mutation rates and fitness effects, while also accounting for unmarked CNVs. However, the model does not account for population heterogeneity of CNVs and their fitness effects. Despite these limitations of distinguishing mutation rate and fitness differences, the authors' conclusions are well supported in that the LTR and ARS deletions have a clear impact on the CNV mediated evolutionary outcome and the mechanism of CNV formation.

---

## [Referee Report · Reviewer #4 (Public review)]

Summary:

Various 'simple' models are used to mechanistically explain the formation of genomic rearrangements, often based on local sequence elements. Here the authors show these models to be lacking for the well characterised GAP1 locus as, although predicted events are observed at reasonable frequency, mutating relevant local sequence elements has surprisingly little impact on the emergence of GAP1 CNV. Rather, a similar set of mechanisms occur (at in some cases somewhat lower frequency) using different genomic elements, the outcome being that that CNV frequency is largely independent of local genomic elements, although this does of course strongly influence the actual structure of the CNVs.

Strengths:

This is a very thorough study of a very complex system.

Weaknesses:

There are limitations as previous reviews have noted, but these are well justified in the revised text and rebuttal

---

## [Author Response]

The following is the authors’ response to the original reviews.

**Public Reviews**:
**Reviewer #1 (Public Review):**
Summary:The work by Chuong et al. provides important new insights into the contribution of different molecular mechanisms in the dynamics of CNV formation. It will be of interest to anyone curious about genome architecture and evolution from yeast biologists to cancer researchers studying genome rearrangements.

Thank you for recognizing the broad significance of our study.

Strengths:Their results are especially striking in that the "simplest" mechanism of GAP1 amplification-non-allelic homologous recombination between the flanking Ty-LTR elements is not the most common route taken by the cells, emphasizing the importance of experimentally testing what might seem on the surface to be obvious answers. One of the important developments of their work is the use of their neural network simulation-based inference (nnSBI) model to derive rates of amplicon formation and their fitness effects.

We agree with this assessment as the results of our study challenge our intuition that the simplest path to structural variation is the most likely and reveals the great diversity in mechanisms that can lead to large scale changes in the genome.

Weaknesses:The manuscript reads as though two different people wrote two different sections of the manuscript - an experimental evolutionist and a computational scientist. If the goal is to reach both groups of readers, there needs to be more explanation of both types of work. I found the computational sections to be particularly dense but even the experimental sections need clearer explanations and more specific examples of the rearrangements found. I will point out these areas in the detailed remarks to the authors. While I have no reason to question their conclusions, I couldn't independently verify the results that ODIRA was the majority mechanism since the sequence of amplified clones was not made available during the review. I've encouraged the authors to include specific, detailed sequence information for both ODIRA events as well as the specific clones where GAP1 was amplified but the flanking gene GFP was not.

We have revised the manuscript to expand explanations of both the experimental and computational aspects of our study and to provide additional information for the reader. In doing so, we have edited the text to improve readability. We have made all raw data publicly available through the NCBI short read archive (SRA) and are hosting all sequence data for easy visualization in JBrowse using a public server.

**Reviewer #2 (Public Review):**
Summary:This study examines how local DNA features around the amino acid permease gene GAP1 influence adaptation to glutamine-limited conditions through changes in GAP1 Copy Number Variation (CNV). The study is well motivated by the observation of numerous CNVs documented in many organisms, but difficulty in distinguishing the mechanisms by which they are formed, and whether or how local genomic elements influence their formation. The main finding is convincing and is that a nearby Autonomous Replicating Sequence (ARS) influences the formation of GAP1 CNVs and this is consistent with a predominate mechanism of Origin Dependent Inverted Repeat Amplification (ODIRA). These results along with finding and characterizing other mechanisms of GAP1 CNV formation will be of general interest to those studying CNVs in natural systems, experimental evolution, and in tumor evolution. While the results are limited to a single CNV of interest (GAP1), the carefully controlled experimental design and quantification of CNV formation will provide a useful guide to studying other CNVs and CNVs in other organisms.

Thank you for this positive assessment of our study.

Strengths:The study was designed to examine the effects of two flanking genomic features next to GAP1 on CNV formation and adaptation during experimental evolution. This was accomplished by removing two Long Terminal Repeats (LTRs), removing a downstream ARS, and removing both LTRs and the ARS. Although there was some heterogeneity among replicates, later shown to include the size and breakpoints of the CNV and the presence of an unmarked CNV, both marker-assisted tracking of CNV formation and modeling of CNV rate and fitness effects showed that deletion of the ARS caused a clear difference compared to the control and the LTR deletion.The consequence of deletion of local features (LTR and ARS) was quantified by genome sequencing of adaptive clones to identify the CNV size, copy number and infer the mechanism of CNV formation. This greatly added value to the study as it showed that (i) ODIRA was the most common mechanism but ODIRA is enhanced by a local ARS, (ii) non-allelic homologous recombination (NAHR) is also used but depends on LTRs, and (iii) de novo insertion of transposable elements mediate NAHR in strains with both ARS and LTR deletions. Together, these results show how local features influence the mechanism of CNV formation, but also how alternative mechanisms can substitute when primary ones are unavailable.

We agree with this assessment.

Weaknesses:The CNV mutation rate and its effect on fitness are hard to disentangle. The frequency of the amplified GFP provides information about mutation rate differences as well as fitness differences. The data and analysis show that each evolved population has multiple GAP1 CNV lineages within it, with some being unmarked by GFP. Thus, estimates of CNV fitness are more of a composite view of all CNV amplifications increasing in frequency during adaptation. Another unknown but potential complication is whether the local (ARS, LTR) deletions influence GAP1 expression and thus the fitness gain of GAP1 CNVs. The neural network simulation-based inference does a good job at estimating both mutation rates and fitness effects, while also accounting for unmarked CNVs. However, the model does not account for the population heterogeneity of CNVs and their fitness effects. Despite these limitations of distinguishing mutation rate and fitness differences, the authors' conclusions are well supported in that the LTR and ARS deletions have a clear impact on the CNV-mediated evolutionary outcome and the mechanism of CNV formation.

While it is true that the inferred mutation rate and fitness effect are negatively correlated, as in other studies (Gitschlag et al., 2023; Caspi et al., 2023; Avecilla et al., 2022), our modeling approach does generate an estimate of each parameter that is best explained by the data. By reporting the confidence intervals (i.e. the 95% HDI) we define the set of parameter values that are consistent with the data. It is true that our model doesn't explicitly account for population heterogeneity; rather, following Hegreness et al. (2006), we employ a single effective fitness effect and mutation rate for all GAP1 CNVs. It is interesting to consider whether the ARS and LTR affect *GAP1* expression; however, we have no evidence that this is the case.

**Reviewer #3 (Public Review):**
Summary:The authors represent an elegant and detailed investigation into the role of cis-elements, and therefore the underlying mechanisms, in gene dosage increase. Their most significant finding is that in their system copy number increase frequently occurs by what they call replication errors that result from the origin of replication firing.The authors somewhat quantitatively determine the effect of the presence of a proximal origin of replication or LTR on the different CNV scenarios.Strengths:(1) A clever and elegant experimental design.(2) A quantitative determination of the effect of a proximal origin of replication or LTR on the different CNV scenarios. Measuring directly the contribution of two competing elements.(3) ODIRA can occur by firing of a distal ARS element.(4) Re-insertion of Ty elements is interesting.

We agree that these are interesting and novel findings from our study.

Weaknesses:(1) Overall, the research does not considerably advance the current knowledge. The research does not investigate what the maximum distance between ARS for ODIRA is to occur. This is an important point since ODIRA was previously described. A considerable contribution to the field would be to understand under what conditions ODIRA wins NAHR.

We agree that these are important questions and they are ones that we are pursuing in future studies.

(2) The title and some sentences in the abstract give a wrong impression of the generality and the novelty of the observations presented. Below are some examples of much earlier work that dealt with mechanisms of CNV and got different conclusions. The Lobachev lab (Cell 2006) published a different scenario years ago, with a very different mechanism (hair-pin capped breaks). The Argueso lab found something different (NAHR) (Genetics 2013).In fact, the CUP1 system presents a good example of this point. The Houseley group showed a complex replication transcription-based mechanism (NAR 2022, cited), the Argueso group showed Ty-based amplification and the Resnick group showed aneuploidy-based amplification. While aneuploidy is a minor factor here the numerous works in Candida albicans, Cryptococcus neoformans, and Yeast suggest otherwise (Selmecki et al Science 2006, Yona et al PNAS 2013, Yang et al Microbiology Spectrum 2021).

As the reviewer points out there have been several important published studies investigating mechanisms by which structural variation is generated. It is important to note that we are explicitly looking at CNVs in the context of adaptive evolution and the role of genomic features that enable different mechanisms of CNV formation. To emphasize this point, we have changed the title of our manuscript to “Template switching during DNA replication is a prevalent source of adaptive gene amplification”. Aneuploidy is indeed a mechanism of adaptive gene amplification in our current and previously reported studies. We have expanded our discussion to place our study in the context of previous studies reporting mechanisms of gene amplification.

(3) The authors added a mathematical model to their experimental data. For me, it was very difficult to understand the contribution of the model to the research. I anticipated, for example, that the model would make predictions that would be tested experimentally. For example, " ARSΔ and ALLΔ are predicted to be almost eliminated by generation 116, as the average predicted WT proportion is 0.998 and 0.999" But to my understanding without testing the model.

In our previous publication (Avecilla et al. 2022, *PLoS Biology*) we experimentally validated the use of nnSBI to infer evolutionary parameters. In this study, we have extended our modeling framework to quantify differences between genotypes, which was not previously possible. Our results reveal that the local ARS has a key role in the overall supply rate of CNVs at this locus.

**Recommendations for the authors**:

We have addressed all public reviews and recommendations.

**Reviewer #1 (Recommendations For The Authors):**
Specific comments about the work are covered in the order of appearance in the text or Figures. I apologize in advance for the number of comments. They are made out of curiosity, enthusiasm for the research, and a desire to help highlight the most interesting aspects of this work.

We are grateful for the thoughtful comments that have helped us to significantly improve our manuscript.

(1) I would appreciate the inclusion of several references to the work on the ODIRA model.a) Page 3 last paragraph: "(2) DNA replication-based mechanisms (Harel et al., 2015; Hastings, Lupski, et al., 2009; Malhotra & Sebat, 2012; Pös et al., 2021; Zhang, Gu, et al., 2009; Brewer et al., 2011)" (Addition of Brewer et al., 2011).

We have added all suggested references.

b) Page 4 top: (Brewer et al., 2011; Brewer et al., 2015; Martin et al., 2024). (Addition of Brewer et al., 2011).

We have added all suggested references.

c) Page 14 top: "Recent work has proposed that ODIRA CNVs are a major mechanism of CNVs in human genomes (Brewer et al., 2015; Martin et al., 2024; Brewer et al., 2024)." Brewer et al., 2024 focuses specifically on ODIRA and human CNVs. (Addition of Brewer et al., 2024).

We have added all suggested references.

(2) Page 6, third paragraph: I was surprised that a single inoculating strain was used to establish the replicate chemostats because of the possibility of non-independence of the resulting GAP1 CNVs. A nnSBI model was used to correct for this possibility later in the paper. It seems like it could have been avoided by a simple change in protocol to inoculate each chemostat with an independent inoculum. Was there a reason that the replicate chemostats were not conducted as independent events? Establishing the presence of 'founder' GAP1 CNVs without GFP seems rather secondary to the point of the paper (examining the CNVs that arise during evolution) and I would recommend it being moved to the supplement.

As is typical in microbial experimental evolution studies, we aimed to start with genetically identical homogenous populations and observe the emergence and selection of de novo variation. Therefore, we founded independent populations from a single inoculum. However, this study, and our prior work using lineage tracking barcodes, has clearly demonstrated that during the initial growth of the culture used for the inoculum CNVs are generated that contribute to the adaptation dynamics on all derived populations. This unanticipated result now suggests that the reviewer’s suggestion is a valid one - independent populations should be derived from independent inocula and this will be our standard practice in future studies.

We believe that our results, presented in Figure 2, establishing the presence of pre-existing GAP1 CNVs without the GFP are important as it highlights a limitation of the use of CNV reporters of gene copy number that was not previously known. However, we subsequently show that this class of variant - CNVs that are not detected by the reporter system - can be incorporated into our modeling framework enabling estimation of evolutionary parameters, which we believe is an important finding warranting inclusion in the main text.

(3) Page 7 first full paragraph: "Finally, we also observe a significant delay (ANOVA, p = 0.00833) in the generation at which the CNV frequency reaches equilibrium in ARS∆ (~generation 112) compared to WT (pairwise t-test, adjusted p = 0.05) . . .". Is the delay in reaching a plateau in Figure 1E just a consequence of the later appearance of CNVs or do the authors believe there are two separate events responsible for this delay? E.g. if the authors think that the delay in reaching a plateau is related to lower selection coefficients of the CNVs that do arise compared to the CNVs of other strains, then this should be explicitly discussed.

We believe that the delay in reaching equilibrium is a consequence of both a lower CNV formation and reduced selection coefficients. Lower values for the fitness coefficient and formation rate in ARS∆ explain both the delay in CNV appearance and CNV equilibrium as shown by the predicted dynamics (Figure S3B). We have added an explicit discussion of the effect of the ARS on CNV dynamics in paragraph 2 of the Discussion section paragraph 2 starting at line 456.

(4) Page 7: Incorporating pre-existing CNVs into an evolutionary model: The rationale for how you are able to discount the formation rate of GFP-free CNVs (C-) in your model isn't clear to me. How are you able to assume that these C- events don't form after timepoint 0? Why do you assume a starting population of C- events but not a starting population of C+ events?

We explored the possibility of modeling C- (amplifications of GAP1 without amplification of the reporter) during the evolution experiment. However, because the rate at which C- events occurs is slower than the rate at which C+ events occur (GAP1 amplifications with amplification of the reporter) we found that the effect was negligible. Importantly, the simple model is sufficient to describe the observed dynamics and thus we do not include these possible rare events.

(5) Figure 1:(a) Panel B: Please put the tRNAs on the line diagrams of the four strains. I first interpreted ALLΔ as missing the tRNAs, too.

Thank you for this suggestion. We added tRNAs to all diagrams to provide additional detail about the structure of the *GAP1* locus.

(b) Panels C, D, and E: the dark shade of the colored boxplots obscures the individual points. I recommend reducing the opacity of the box or choosing a lighter shade so that the individual points are visible on top of the box. Is the percent increase in CNVs per generation (Panel D) based on the slopes of the curves in panel B? By eye the slopes of ARS∆ and ALL∆ appear at least as steep as those of wild type and LTR∆.

Thank you for this suggestion. We have now made the individual points visible on top of the boxplots in Figures 1C, 1D, and 1E. The lines in Figure 1B show the median value across populations per time point whereas each point in Figure 1D is the slope from linear regression using values from individual populations (data from individual populations are shown in Figure 3C).

(6) Figure 2:(a) Panel A: Please remind the readers what FSC-A is measuring and label the different groups of cells in each sample. Are we supposed to assume the upper scatter in generation 8 is the pre-existing CNV variants? Are the three species at generation 50 due to 1, 2, and 3 copies of GFP? Is the new species in generation 137 further amplification of the locus? And if so, how many copies does it represent? I find it fascinating that what I assume is the 2-copy CNV (presumably a direct oriented amplicon produced by NAHR) at 50 generations is lost (out-competed by a potential inverted triplication) at later times, but I didn't find any mention of this phenomenon in the text. What do the different mutant strains look like over the same time course? Please supply supplemental figures with the flow cytometry gating and vertically aligned histograms of the GFP signal so that the peaks are more easily compared. And provide this information for each of the altered strains in supplementary materials.

Thank you for these useful suggestions. We have added a gating legend to the figure to clearly indicate the copy-number for each subpopulation. We have edited the caption and main text to explain forward scatter (FSC-A). Raw flow cytometry plots are now provided as Supplementary figure 2 and distributions of cell-size normalized GFP signal are provided in Supplementary figure 3. Although our primary objective with Figure 2A was to show the persistence of the 1-copy GFP population the reviewer is correct that we did not highlight interesting aspects of the CNV dynamics. We have added additional text starting at line 251 to point out these features of the data.

(b) Panel B: It would help to label the different colored boxes inside cells in Figure 2B - it took me a while to identify the white box as an unrelated adaptive mutation elsewhere in the genome. The linear arrangement of these small colored blocks seems to indicate their structural arrangement. Is that the case? And are they inverted or direct amplicons? Perhaps the authors are being agnostic at this point but it would be better if each of the blocks were separate. If there are other mutations that can explain these GFP-non-amplified survivors, were they identified in your whole genome sequencing?

We have now included a complete legend for Figure 2B indicating that the white box reflects other beneficial mutations. We have separated this class of beneficial mutation from the GAP1 and reporter elements to reflect that they are not linked. We did not identify additional beneficial mutations but plan to pursue this question in a future project.

(c) Panel C: Are the two sets of lines mislabeled? One would expect the "reported" CNV proportions to be lower than the total CNV proportions, not the other way around. Maybe the labels "total CNVs" and "reported CNVs" are unclear to me and I am misunderstanding what "reported" refers to. Please clarify.

Thank you for identifying this mistake. The lines were mislabeled and have now been corrected in the revised version.

(7) Figure 3:(a) A fuller discussion of panels A and B is needed. The results of panel A in particular seem like an excellent opportunity for connecting the computation to the biology. Can the authors speculate on why the ALL∆ strain has a higher CNV formation rate (𝛿c) than the ARS∆ strain? I would think that taking away one means of amplification would decrease CNV formation. Likewise, could the authors discuss why the selection coefficient (sc) for the LTR∆ strain would be the same as for the wild type? Overall, I would like to see more discussion about what these differences in formation rates and selection coefficients could mean for the types of amplicons arising in the chemostats. (In panel B I don't see the shaded area referred to in the figure legend.) A side-by-side comparison of the data in Panel A with the data shown in Supplemental Figure S3A would be instructive..

Thank you for raising these points. We have added substantial text to the manuscript to address these findings. Starting at line 456 we state:

“The lower CNV formation rate in the LTR∆ could be a closer approximation of ODIRA formation rates at this locus as ODIRA CNVs are the predominant CNV mechanism in the LTR∆ strain (Figure 4F). Furthermore, the low formation rates in the LTR∆ relative to WT might suggest that the presence of the flanking long terminal repeats may increase the rate of ODIRA formation through an otherwise unknown combinatorial effect of DNA replication across these flanking LTRs and template switching at the GAP1 locus. ARS∆ has the lowest CNV formation rate and it could be an approximation of the rates of NAHR between flanking LTRs and ODIRA at distal origins. We find that the ALL∆ has a higher CNV formation rate than the ARS∆, even though three elements are deleted instead of one. One explanation for this is that the deletion of the flanking LTRs in ALL∆ gives opportunity for novel transposon insertions and subsequent LTR NAHR. Indeed we find an enrichment of novel transposon-insertions in the ALL∆ (Figure 4F) and subsequent CNV formation through recombination of the Ty1-associated repeats (Figure 4H, ALL∆). Both events, transposon insertion followed by LTR NAHR, would have to occur quickly at a rate that explains our estimated CNV rate in ALL∆. While remarkable, increased transposon activity has been associated with nutrient stress (Curcio & Garfinkel, 1999; Lesage & Todeschini, 2005; Todeschini et al., 2005) and therefore feasible explanation for the CNV rate estimated in the ALL∆. Additionally, ARS∆ clones rely more on LTR NAHR to form CNVs (Figure 4F). The prevalence of ODIRA in ARS∆ and ALL∆ are similar. LTR NAHR usually occurs after double strand breaks at the long terminal repeats to give rise to CNVs (Argueso et al., 2008). Because we use haploid cells, such double strand break and homology-mediated repair would have to occur during S-phase after DNA replication with a sister chromatid repair template to form tandem duplications. Therefore the dependency on LTR NAHR to form CNVs and the spatial (breaks at LTR sequences) and temporal (S-phase) constraints could explain the lower formation rate in ARS∆.”

In addition, we added a discussion of the different selection coefficients estimated and how the simulated competitions help us understand the decreased selection coefficients in the architecture mutants. In newly added text starting at line 479 we state:

“The genomic elements have clear effects on the evolutionary dynamics in simulated competitive fitness experiments. The similar selection coefficients in WT and LTR∆ suggest that CNV clones formed in these background strains are similar. Indeed, the predominant CNV mechanism in both is ODIRA followed by LTR NAHR (Figure 4F). While LTR NAHR is abolished in the LTR∆, it seems that CNVs formed by ODIRA allow adaptation to glutamine-limitation similar to WT. The lower selection coefficients in ARS∆ and ALL∆ suggest that GAP1 CNVs formed in these strains have some cost. In a competition, they would get outcompeted by CNV alleles in the WT and LTR∆ background.”

(b) The data shown in panel C seems redundant to what is shown more clearly in Supplemental Figure S3B. It seems to me the more important comparison to make in panel C would be the overlay of the predicted data to the median proportion of cells obtained from the experimental data (Figure 1B). Also, overlays of the cultures from each strain could be added to S3A. It is difficult to see the variation within each strain when the data from all four strains are superimposed as they are in Figure 3C.

We agree and have edited Figure 3C to incorporate these suggestions and more clearly convey the intra- and interstrain variation.

(8) Figure 4:(a) Panels A, B, and C are nice summaries and certainly helpful for understanding panel E, but it would be instructive to see some actual rearrangements of the ODIRA events, the NAHR, and the transposon-mediated rearrangements. It isn't clear to me what these last events look like. A figure that shows the specific architecture of example clones for each category would be helpful. I am also having a hard time reconciling ODIRA events with a copy number of 2. Are these rearrangements free isochromosomes with amplification to the telomere or are they secondary rearrangements like those described in Brewer et al., 2024? And what about the non-aneuploid rearrangement that includes the centromere? Is it a dicentric?

We have now added more detailed depictions of CNVs in Figure 4A and provide links to visualize the alignment files. We have added additional discussion starting at line 397 of the non-canonical ODIRA events and putative neochromosome amplicons with reference to Brewer et al 2024. Starting at line 397 we state:

“Surprisingly, we found CNVs with breakpoints consistent with ODIRA that contained only 2 copies of the amplified region, whereas ODIRA typically generates a triplication. In the absence of additional data, we cannot rule out inaccuracy in our read-depth estimates of copy numbers for these clones (ie. they have 3 copies). An alternate explanation is a secondary rearrangement of an original inverted triplication resulting in a duplication (Brewer et al., 2024); however, we did not detect evidence for secondary rearrangements in the sequencing data. A third alternate explanation is that a duplication was formed by hairpin capped double-strand break repair (Narayanan et al., 2006). Notably, we found 3 additional ODIRA clones that end in native telomeres, each of which had amplified 3 copies. In these clones the other breakpoint contains the centromere, indicating the entire right arm of chromosome XI was amplified 3 times via ODIRA, each generating supernumerary chromosomes. Thus,ODIRA can result in amplifications of large genomics regions from segmental amplifications to supernumerary chromosomes.”

(b) In Panel B the violin plots appear to indicate that there are two size categories for amplicons in the ARS∆ strain. Do clones from these different sub-populations share a common CNV architecture?

Thank you for making this point. (Please note that the violin plots are now Figure 4E) We added a short discussion and Supplementary Figure 14. In line 432, we state:

“In ARS∆, we find two CNV length groups (Figure 4E) that correspond with two different CNV mechanisms (Supplementary Figure 14). 100% of smaller CNVs (6-8kb) (Supplementary Figure 14) correspond with a mechanism of NAHR between LTRs flanking the GAP1 gene (Figure 4H, ARS∆, bottom left green points). Larger CNVs (8kb-200kb) (Supplementary Figure 14) correspond with other mechanisms that tend to produce larger CNVs, including ODIRA and NAHR between one local and one distal LTR element (Figure 4H).”

(c) Panels D and E: There is great information in these two panels but I find the color keys confusing. There doesn't seem to be any reason for the strain color key in panel E. I am assuming that the key should go with Panel D. Is there some way to indicate in Panel D which events are in which CNV category? It is cumbersome to find that information from Panel E. Perhaps the color-coding from Panel E could be applied to the row labels in Panel D. Being able to link amplicon to the mechanism of CNV formation is especially important for seeing which ODIRA events contain an origin.

Thank you for this suggestions. We now indicate the mechanism of CNV formation using a consistent color coding in panels G and H (previously panels D and E).

(d) Panel E: I don't understand the two axes in Panel E. If both axes are log scales, why is the origin 0 for the X-axis and 1 for the Y-axis? And why are the focal amplicons (most of which are recombination events between the two LTRs) scattered in both X and Y coordinates? Shouldn't they form a single point? The same for the recombinants with distal LTRs. Also, orange and red (ODIRA and complex CNVs, respectively) are very hard to distinguish. All of these data need to be presented in a spreadsheet identifying each clone's strain ID, chemostat number, GAP1 and GFP copy numbers, sequence across the junction, and their coordinates. The SRA project (PRJNA1016460) for the sequence data was not found in SRA. Will this data be available to easily look at read depth across chromosome XI for all of the sequenced strains - perhaps as .bam files?

Thank you for calling these issues with data visualization to our attention. Indeed, the focal amplifications do form around a single point. We originally had jittered the data to show each individual focal amplification but agree that this is confusing. We now overlay the individual points and have altered opacity to enable visualization of individual values. The suggested table of clone data is provided in Supplementary File 2 and the SRA project is now publicly available. Moreover, we are providing all alignment (.bam) files, split, and discordant read depth profiles for each CNV strain and their corresponding ancestor aligned to our custom reference genomes in a public jbrowse server at:

https://jbrowse.bio.nyu.edu/gresham/?data=data/ee_gap1_arch_muts for WT strains, https://jbrowse.bio.nyu.edu/gresham/LTRKO_clones for LTR∆ strains, https://jbrowse.bio.nyu.edu/gresham/ARSKO_clones for ARS∆ strains, https://jbrowse.bio.nyu.edu/gresham/ALLKO_clones for ALL∆ strains.

(e) Supplementary Table 1 and Supplementary Figure S2: Please indicate which rearrangements (of the 8 reported in Figure S2A) were identified in each of the clones described in the table. If each of the 8 amplicons is identified by a letter, then this information could be added as a column in the table. I am assuming that each of the eight rearrangements was found in more than one chemostat. Showing these data is crucial for establishing the possibility that they were preexisting at the time of chemostat inoculation. The other possibility is that the clones with amplified GAP1 but a single copy of GFP could have been created by a secondary rearrangement in the outgrowth of the clones that originally had amplified both genes to the same extent. What is the structure of these amplicons? Is there a common junction between GAP1 and GFP? I couldn't find these data in the paper. A suggestion for Supplemental Figure S2A - include a zoomed-in inset for the GAP1 GFP region for each of the 8 read-depth plots. It is hard to see the exact location of GFP and GAP1 across all 8 tracks without getting out a ruler. Were these sequences aligned to your custom reference genome or the reference genome without GFP? If they were aligned to the custom reference that includes the GFP reporter, the reader could visually confirm the absence of GFP amplification.

Thank you for these suggestions. We edited Supplementary Table 1 and Supplementary Figure 1A as requested. We now provide the precise CNV breakpoints in the GFP-GAP1 region (supplemental figure 1B) displaying both genome read depth and split read depth tracks. These sequences were aligned to the custom reference containing the GFP reporter, which is now clearer in the figure and caption text in line 1226.

The clones in this figure were sampled from the five different chemostats and we have clarified this in the edited table and text at line 210. We did not detect the same CNV allele in different chemostats and therefore we do not have evidence to support *GAP1* amplification without the GFP reporter pre-existing at time of inoculation. We are not able to definitively distinguish whether the amplicons were pre-existing at the time of inoculation or occurred after as we do not have barcoded lineages. We isolated clones carrying this class of amplification from the 1-GFP-copy subfraction late in the experimental evolution (generation 165-182). Given that the alleles appear to differ between populations we think the most parsimonious explanation is that these amplifications occurred after chemostat inoculation but early in the evolution experiment. We explicitly state this in the text starting in line 219.

(9) Page 8-9: I am sorry to say that I can't evaluate the "HDI of posterior distributions". It is out of my competency range. So I am not sure what this analysis is adding to the paper. The same goes for the rest of the supplementary figures.

HDI is a measure of certainty in an estimate, similar to confidence interval. We state this in the text in line 276. With the editing of the text we hope the modeling and its supplementary figures are more clear now.

(10) Page 9 top: Deletion of the ARS appears to lower the fitness of the amplified GAP1 variants. Can the authors speculate on why the ARS deletion would reduce fitness? Did they consult published replication profiles to determine the size of the origin-free gap that could result from the deletion of this mid-S phase origin? Could it explain the delay in the appearance of GAP1 amplicons in the ARS-deletion strains and be responsible for their reduced selection coefficients? Did you examine the growth properties of the starting strain or any of the amplified GAP1 derivatives? Perhaps this consideration could contribute to the discussion. Could there be a bit fuller discussion on the interaction between CNV length differences as shown in Figure 4A and differences in selection coefficient as determined by the nnSBI?

Thank you for raising this point. We have now added text to our discussion of the reduced fitness in ARS∆ in relation to DNA replication starting on line 359:

“ARS1116 is a major origin (McGuffee et al., 2013) and ODIRA CNVs found around this origin corroborate its activity. GAP1 is highly transcribed in glutamine-limited chemostats (Airoldi et al., 2016). Head-on transcription-replication collisions at this locus may be contributing to the higher CNV formation rate in wild type and LTR∆. Elimination of the local ARS could result in less transcription-replication collisions and the slower CNV formation rates estimated. Once formed they get outcompeted by faster-forming CNVs and thus in theory are less fit than CNVs in other strain backgrounds. These simulated competitions further suggest that the ARS is a more important contributor to adaptive evolution mediated by GAP1 CNVs.”

We examined replication profiles in McGuffee et al. Mol Cell. 2013 but could not determine the size of the origin-free gap. ARS1116 and its neighboring ARSs, ARS1118 downstream and ARS1115 upstream are efficient firing origins (Supplement 1 of McGuffee et al. 2013) and therefore the gap is likely to be minimal. The dynamics of the distal firing ARS elements involved in creating ODIRA CNVs might explain the reduced fitness, but further experiments would be required to address this. Regarding growth properties, the growth rate at steady-state in the chemostat is the same as the dilution rate regardless of strain background. Because we had the same dilution rate for each chemostat, the ARS∆ populations would have the same replication rate as the other three strains even if there may be replication rate differences in bulk culture growth. Finally, we found no significant interaction between CNV length and selection coefficients and we state this in line 359.

(11) Page 10: WT competition simulations: It may help to explicitly state that the competition modeling approach was experimentally validated in Avecilla 2022 as opposed to just citing the paper. I found the results much more convincing after reading Avecilla 2022, but I imagine many readers may skip that.

We added a sentence to state that the nnSBI method was experimentally validated in Avecilla et 2022 at line 249.

**Reviewer #2 (Recommendations For The Authors):**
(1) Figure 2: says reported CNV proportions (dashed). This may be a typo since I think the GFP reported should be solid, not dashed. Also, (C) isn't bold.

Thank you for identifying these mistakes. We have corrected the figure’s caption in line 1157.

(2) "compared to 898/345 clones" Does this refer to transposition/clone? Seems more natural to compare clones with transpositions to a total number of clones. This could be clarified.

We rephrased the sentence (lines 519-520) to clarify that in their study Hays et al. 2023 found 898 novel Ty insertions across 345 nitrogen-evolved clones. As a result of this high rate of transposition, some clones are expected to have multiple Ty insertions.

(3) The methods state that Kan replaces the Nat cassette that was used to make the deletions. It should be made more clear whether Kan is present and where Kan is with respect to GFP and GAP1.

Thank you for pointing this out. To clarify we added the following sentence to the methods starting in line 567:

“The CNV reporter is 3.1 kb and located 1117 nucleotides upstream of the GAP1 coding sequence. It consists of, in the following order, an ACT1 promoter, mCitrine (GFP) coding sequence, ADH1 terminator, and kanamycin cassette under control of a TEF promoter and terminator.”

Additionally in line 571 we clarify the drug resistance of the genomic architecture ∆ strains that are kanamycin(+) and nourseothricin(-).

**Reviewer #3 (Recommendations For The Authors):**
(1) The major advancement of the manuscript is stated in the title "DNA replication errors are a major source of adaptive gene amplification" First, in my humble opinion the term replication errors is not quite right; the term template switching is more accurate. In that regard, recently a paper was published just on this topic (Martin et al Plos Genetics, 2024).

We have changed the title to “Template-switching during DNA replication is a prevalent source of adaptive gene amplification”. We cite Martin et al Plos Genetics 2024 throughout the main text in lines 93, 126, 159, 502, 555.

(2) I find the statement "We find that 49% of all GAP1 CNVs are mediated by the DNA replication-based mechanism Origin Dependent Inverted Repeat Amplification (ODIRA) regardless of background strain." Somewhat misleading, there were considerable differences between the strains. If I am not mistaken the range was 20-80%.

Thank you for pointing this out. Indeed, the range was 26-80% across the four strains. We updated this sentence in the abstract at line 40, and in the main text at line 141 to clearly state the range.

(3) In their attempt to fill the gap of knowledge regarding the fitness effect of the adaptive CNV the authors use a mathematical model. As an experimental biologist, I found the description lacking. It is hard for me to evaluate the contribution of the model to understanding the results and I think the authors could improve this part.

We have edited the text regarding the modeling and associated results and hope that it is now more clear. The mathematical model describes the experiment in a simplified manner. We use it to predict the outcomes of additional experiments without additional experimental work. For example, we used it to simulate a competition between two strains, predict the total proportion of GAP1 CNVs, and predict the relative genetic diversity.

(4) Experiments the authors may want to consider to increase the novelty of their work:a) Place the GAP1 gene right in the middle of the two most distant ARS elements and test the mechanism of CNV.

Thank you for this proposed experiment. It is beyond the scope of this paper and will be pursued in future studies.

b) The finding of de-novo Ty element insertion is interesting. What happens if the overdose strain of Jef Boeke is used (Retrotransposon overdose and genome integrity, PNAS 2009) or in contrast, a reverse transcriptase deficient strain?

We agree. Our study has revealed a critical role for novel Ty insertion in mediating CNVs. The suggested experiments as well as using strains that lack Ty sequences will be very interesting to explore in followup studies.

c) The genomic analyses were based on single colony isolates. To my understanding, the CNV events are identified at least partly by split reads. Therefore, each event may have a "signature" that is unique and can be concluded from single reads and not necessarily from the assembled genome. If true, a distinction between the scenarios could be achieved if bulk cultures are sequenced with enough depth. Thus, a truly dynamic and quantitative determination of the different events, rate of appearance, and disappearance can be made.

Thank you for this suggestion, which is a good idea but not currently feasible for several reasons. First, although split reads are a powerful way to detect CNV breakpoints, we have found that even at high coverage (21-153X, median 78.5X), in clonal samples that are rare with only 3-30 split reads (median 14) detected. These observations are from a total of 23 breakpoints across 16 sequenced clones. Thus, when sequencing heterogeneous cultures, in which different CNVs only comprise a fraction of the population, our ability to detect single CNV alleles by split reads and quantify their frequency is limited. Given our observations, with a median of 14 split reads when sequencing to 78.5X genome-wide read coverage it is possible we may be able to detect an individual CNV allele once it makes up (14/78.5) 17% of the population. However, our previous study has shown that there are tens to hundreds of unique CNV alleles initially and thus this would only be feasible at very late timepoints. Second, recurrent CNVs may occur independently at the same exact location, such as LTR NAHR. Thus, unique signatures may not be obtained even if they are independent events. Third, it would be not appropriate to pursue this analysis with our current dataset, as we lack lineage tracking barcodes to validate the results.